# Pareto Frontiers in Deep Feature Learning: Data, Compute, Width, and Luck

**Benjamin L. Edelman**[1]    **Surbhi Goel**[2]    **Sham Kakade**[1]    **Eran Malach**[3]    **Cyril Zhang**[4]

[1]Harvard University    [2]University of Pennsylvania
[3]Hebrew University of Jerusalem    [4]Microsoft Research NYC

bedelman@g.harvard.edu, surbhig@cis.upenn.edu, sham@seas.harvard.edu,
eran.malach@mail.huji.ac.il, cyrilzhang@microsoft.com

## Abstract

In modern deep learning, algorithmic choices (such as width, depth, and learning rate) are known to modulate nuanced resource tradeoffs. This work investigates how these complexities necessarily arise for feature learning in the presence of computational-statistical gaps. We begin by considering offline sparse parity learning, a supervised classification problem which admits a statistical query lower bound for gradient-based training of a multilayer perceptron. This lower bound can be interpreted as a *multi-resource tradeoff frontier*: successful learning can only occur if one is sufficiently rich (large model), knowledgeable (large dataset), patient (many training iterations), or lucky (many random guesses). We show, theoretically and experimentally, that sparse initialization and increasing network width yield significant improvements in sample efficiency in this setting. Here, width plays the role of parallel search: it amplifies the probability of finding "lottery ticket" neurons, which learn sparse features more sample-efficiently. Finally, we show that the synthetic sparse parity task can be useful as a proxy for real problems requiring axis-aligned feature learning. We demonstrate improved sample efficiency on tabular classification benchmarks by using wide, sparsely-initialized MLP models; these networks sometimes outperform tuned random forests.

## 1    Introduction

Algorithm design in deep learning can appear to be more like "hacking" than an engineering practice. Numerous architectural choices and training heuristics can affect various performance criteria and resource costs in unpredictable ways. Moreover, it is understood that these multifarious hyperparameters all interact with each other; as a result, the task of finding the "best" deep learning algorithm for a particular scenario is foremost empirically-driven. When this delicate balance of considerations is achieved (i.e. when deep learning works well), learning is enabled by phenomena which cannot be explained by statistics or optimization in isolation. It is natural to ask: *is this heterogeneity of methods and mechanisms necessary?*

This work studies a single synthetic binary classification task in which the above complications are recognizable, and, in fact, *provable*. This is the problem of offline (i.e. small-sample) sparse parity learning: identify a $k$-way multiplicative interaction between $n$ Boolean variables, given $m$ random examples. We begin by interpreting the standard statistical query lower bound for this problem as a *multi-resource tradeoff frontier* for deep learning, balancing between the heterogeneous resources of dataset size, network size, number of iterations, and success probability. We show that in different regimes of simultaneous resource constraints (data, parallel computation, sequential computation,

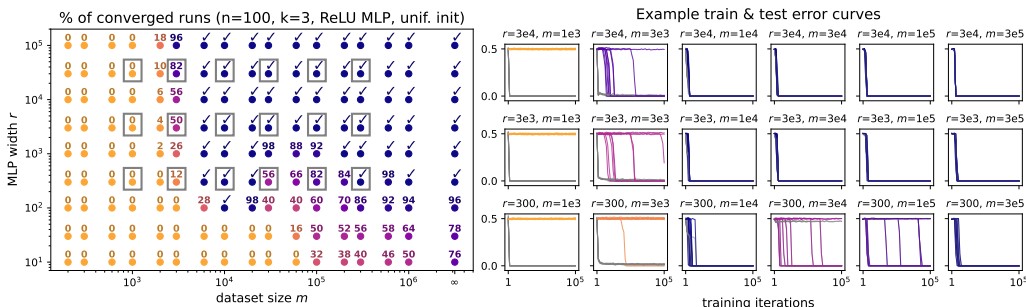

Figure 1: **"More data or larger network?"** Effects of jointly scaling the dataset and model sizes, for 2-layer MLPs trained to learn sparse parities. *Left:* A success frontier, where the computational challenge of feature learning can be surmounted by increasing sample size *or* model size. Each point shows the percentage of successful training runs (out of 50 trials) for each (dataset size $m$, width $r$); (✓) denotes 100% successful trials. *Right:* 10 example training curves (gray = training error; colored = test error) for each of the boxed $(m, r)$ pairs. See main text and Section 4 for further details.

and random trials), the standard algorithmic choices in deep learning can succeed by diverse and entangled mechanisms. Specifically, our contributions are as follows:

**Multi-resource lower and upper bounds.** We formulate a "data × width × time × luck" lower bound for offline sparse parity learning with feedforward neural nets (Theorem 3). This barrier arises from the classic statistical query (SQ) lower bound for this problem. We show that under different resource constraints, the tractability of learning can be "bought" with varied mixtures of these resources. In particular, in Theorems 4 and 5 we prove that by tuning the width and initialization scheme, we can populate this frontier with a spectrum of successful models ranging from narrow networks requiring many training samples to sample-efficient networks requiring many neurons, as summarized by the following informal theorem statement:

**Informal Theorem 1.** *Consider the problem of learning $(n, k)$-parities from $m$ i.i.d. samples with a 2-layer width-$r$ ReLU MLP, whose first-layer neurons are initialized with sparsity $s$. After $O_n(1)$ steps of gradient descent, the $k$ relevant coordinates are identified with probability 0.99 when (1) $s > \Omega(k)$, $r = \Theta((n/s)^k)$ and $m = \Theta(n^2(s/k)^{k-1})$, and when (2) $s < k$, $r = \Theta((n/k)^s)$ and $m = \Theta((n/k)^{k-s-1})$.*

Intuitively, this analysis reveals a feature learning mechanism by which overparameterization (i.e. large network width) plays a role of parallel search over randomized subnetworks. Each individual hidden-layer neuron has its own sample complexity for identifying the relevant coordinates, based on its *Fourier gap* (Barak et al., 2022) at initialization. Trained with parallel gradient updates, the full network implicitly acts as an ensemble model over these neurons, whose overall sample complexity is determined by the "winning lottery tickets" (Frankle and Carbin, 2018) (i.e. the lucky neurons initialized to have the lowest sample complexity). This departs significantly from the *neural tangent kernel* (Jacot et al., 2018) regime of function approximation with wide networks, in which overparameterization *removes* data-dependent feature selection (rather than parallelizing it across subnetworks).

**Empirical study of neural nets' statistical thresholds for sparse parity learning.** We corroborate the theoretical analysis with a systematic empirical study of offline sparse parity learning using SGD on MLPs, demonstrating some of the (perhaps) counterintuitive effects of width, data, and initialization. For example, Figure 1 highlights our empirical investigation into the interactions between data, width, and success probability. The left figure shows the fractions of successful training runs as a function of dataset size (x-axis) and width (y-axis). Roughly, we see a "success frontier", where having a larger width can be traded off with smaller sample sizes. The right figure depicts some training curves (for various widths and sample sizes). Grokking (Power et al., 2022; Liu et al., 2022) (discontinuous and delayed generalization behavior induced by optimization dynamics) is evident in some of these figures.

**"Parity2real" transfer of algorithmic improvements.** It is often observed that deep learning methods underperform tree-based methods (e.g. random forests) on tabular datasets, particularly those where the target function depends on a few of the input features in a potentially non-smooth manner; see Grinsztajn et al. (2022) for a recent discussion. Motivated by our findings in the synthetic parity setting (and the observation that the sparse parity task possesses a number of the same properties of these problematic real-world datasets), we then turn to experimentally determine the extent to which our findings also hold for real tabular data. We evaluate MLPs of various depths and initialization sparsities on 16 tabular classification tasks, which were standardized by Grinsztajn et al. (2022) to compare neural vs. tree-based methods. Figure 3 shows that wider networks and sparse initialization yield improved performance, as in the parity setting. In some cases, our MLPs outperform tuned random forests.

## 1.1 Related work

In the nascent empirical science of large-scale deep learning, *scaling laws* (Kaplan et al., 2020; Hoffmann et al., 2022) have been shown to extrapolate model performance with remarkable consistency, revealing flexible tradeoffs and Pareto frontiers between the heterogeneous resources of data and computation. The present work reveals that in the simple synthetic setting of parity learning, the same intricacies can be studied theoretically and experimentally. In particular, viewing *model size* $\times$ *training iterations* $\times$ *random restarts* as a single *"total FLOPs"* resource, our study explains why data $\times$ compute can be a necessary and sufficient resource, through the lens of SQ complexity.

**Analyses of deep feature learning.** Formally characterizing the representation learning mechanisms of neural networks is a core research program of deep learning theory. Many recent works have analyzed gradient-based feature learning (Wei et al., 2019; Barak et al., 2022; Zhenmei et al., 2022; Abbe et al., 2022; Damian et al., 2022; Telgarsky, 2022), escaping the "lazy" neural tangent kernel (NTK) regime (Jacot et al., 2018; Chizat et al., 2019), in which features are fixed at initialization.

**Learning parities with neural networks.** The XOR function has been studied as an elementary challenging example since the dawn of artificial neural networks (Minsky and Papert, 1969), and has been revisited at various times: e.g. neural cryptography (Rosen-Zvi et al., 2002); learning interactions via hints in the input distribution (Daniely and Malach, 2020); a hard case for self-attention architectures (Hahn, 2020). Closest to this work, Barak et al. (2022) find that in the case of *online* (infinite-data) parity learning, SGD provides a feature learning signal for a *single neuron*, and can thus converge at a near-optimal computational rate for non-overparameterized networks. They note computational-statistical tradeoffs and grokking in the offline setting, which we address systematically. Merrill et al. (2023) examine the same problem setting empirically, investigating a mechanism of competing sparse and dense sub-networks. Abbe et al. (2023) provide evidence that the time complexity required by an MLP to learn an arbitrary sparse Boolean function is governed by the largest "leap" in the *staircase* of its monomials, each of which is a sparse parity. Telgarsky (2022) gives a margin-based analysis of gradient flow on a two-layer neural network that achieves improved sample complexity ($\tilde{O}(n/\epsilon)$) for the 2-sparse parity problem, at the cost of exponential width.

**Neural nets and axis-aligned tabular data.** Decision tree ensembles such as random forests (Breiman, 2001) and XGBoost (Chen and Guestrin, 2016) remain more popular among practitioners than neural networks on tabular data (Kaggle, 2021), despite many recent attempts to design specialized deep learning methods (Borisov et al., 2022). Some of these employ sparse networks (Yang et al., 2022b; Lutz et al., 2022) similar to those considered in our theory and experiments.

We refer the reader to Appendix A for additional related work.

## 2 Background

### 2.1 Parities, and parity learning algorithms

A light bulb is controlled by $k$ out of $n$ binary switches; each of the $k$ influential switches can toggle the light bulb's state from any configuration. The task in parity learning is to identify the subset of $k$ important switches, given access to $m$ i.i.d. uniform samples of the state of all $n$ switches. Formally, for any $1 \leq k \leq n$ and $S \subseteq [n]$ such that $|S| = k$, the parity function $\chi_S : \{\pm 1\}^n \rightarrow \pm 1$ is

defined as $\chi_S(x_{1:n}) := \prod_{i \in S} x_i$.[1] The $(n,k)$-parity learning problem is to identify $S$ from samples $(x \sim \mathrm{Unif}(\{\pm 1\}^n), y = \chi_S(x)))$; i.e. output a classifier with 100% accuracy on this distribution, without prior knowledge of $S$.

This problem has a rich history in theoretical computer science, information theory, and cryptography. There are several pertinent ways to think about the fundamental role of parity learning:

(i) **Monomial basis elements:** A $k$-sparse parity is a degree-$k$ monomial, and thus the analogue of a Fourier coefficient with "frequency" $k$ in the harmonic analysis of Boolean functions (O'Donnell, 2014). Parities form an important basis for polynomial learning algorithms (e.g. Andoni et al. (2014)).

(ii) **Computational hardness:** There is a widely conjectured *computational-statistical gap* for this problem (Applebaum et al., 2009, 2010), which has been proven in restricted models such as SQ (Kearns, 1998) and streaming (Kol et al., 2017) algorithms. The statistical limit is $\Theta(\log(\text{number of possibilities for } S)) = \Theta(k \log n)$ samples, but the amount of computation needed (for an algorithm that can tolerate an $O(1)$ fraction of noise) is believed to scale as $\Omega(n^{ck})$ for some constant $c > 0$ – i.e., there is no significant improvement over trying all subsets.

(iii) **Feature learning:** This setting captures the learning of a concept which depends jointly on multiple attributes of the inputs, where the lower-order interactions (i.e. correlations with degree-$k' < k$ parities) give no information about $S$. Intuitively, samples from the sparse parity distribution look like random noise until the learner has identified the sparse interaction. This notion of feature learning complexity is also captured by the more general *information exponent* (Ben Arous et al., 2021).

## 2.2 Notation for neural networks

For single-output regression, a 2-layer multi-layer perceptron (MLP) is a function class, parameterized by a matrix $W \in \mathbb{R}^{r \times n}$, vectors $v, b \in \mathbb{R}^r$ and scalar $\beta \in \mathbb{R}$, defined by

$$\widehat{y} : x \mapsto v^\top \sigma(Wx + b) + \beta,$$

where $\sigma$ is an activation function, usually a scalar function applied identically to each input. For rows $w_i$ of $W$, the intermediate variables $\sigma(w_i^\top x + b_i)$ are thought of as *hidden layer* activations or *neurons*. The number of parallel neurons is often called the *width*.

## 3 Theory

In this section we theoretically study the interactions between data, width, time, and luck on the sparse parity problem. We begin by rehashing Statistical Query (SQ) lower bounds for parity learning in the context of gradient-based optimization, showing that without sufficient resources sparse parities cannot be learned. Then, we prove upper bounds showing that parity learning is possible with correctly scaling either width (keeping sample size small), sample size (keeping width small), or a mixture of the two.

**Statistical query algorithms.** A seminal work by Kearns (1998) introduced the statistical query (SQ) algorithm framework, which provides a means of analyzing noise-tolerant algorithms. Unlike traditional learning algorithms, SQ algorithms lack access to individual examples but can instead make queries to an *SQ oracle*, which responds with noisy estimates of the queries over the population. Notably, many common learning algorithms, including noisy variants of gradient descent, can be implemented within this framework. While SQ learning offers robust guarantees for learning in the presence of noise, there exist certain problems that are learnable from examples but not efficiently learnable from statistical queries (Blum et al., 1994, 2003). One notable example of such a problem is the parity learning problem, which possesses an SQ lower bound. This lower bound can be leveraged to demonstrate the computational hardness of learning parities with gradient-based algorithms (e.g., Shalev-Shwartz et al. (2017)).

---

[1] Equivalently, parity can be represented as a function of a bit string $b \in \{0,1\}^n$, which computes the XOR of the influential subset of indices $S$: $\chi_S(b_{1:n}) := \oplus_{i \in S} b_i$.

## 3.1 Lower bound: a multi-resource hardness frontier

We show a version of the SQ lower bound in this section. Assume we optimize some model $h_\theta$, where $\theta \in \mathbb{R}^r$ (i.e., $r$ parameters). Let $\ell$ be some loss function satisfying: $\ell'(\hat{y}, y) = -y + \ell_0(\hat{y})$; for example, one can choose $\ell_2(\hat{y}, y) = \frac{1}{2}(y - \hat{y})^2$. Fix some hypothesis $h$, target $f$, a sample $\mathcal{S} \subseteq \{\pm 1\}^n$ and distribution $\mathcal{D}$ over $\{\pm 1\}^n$. We denote the empirical loss by $L_\mathcal{S}(h, f) = \frac{1}{|\mathcal{S}|} \sum_{\mathbf{x} \in \mathcal{S}} \ell(h(\mathbf{x}), f(\mathbf{x}))$ and the population loss by $L_\mathcal{D}(h, f) := \mathbb{E}_{\mathbf{x} \sim \mathcal{D}} [\ell(h(\mathbf{x}), f(\mathbf{x}))]$. We consider SGD updates of the form:

$$\theta_{t+1} = \theta_t - \eta_t \left( \nabla_\theta \left( L_{\mathcal{S}_t}(h_{\theta_t}, f) + R(\theta_t) \right) + \xi_t \right),$$

for some sample $\mathcal{S}_t$, step size $\eta_t$, regularizer $R(\cdot)$, and adversarial noise $\xi_t \in [-\tau, \tau]^r$. For normalization, we assume that $\|\nabla h_\theta(\mathbf{x})\|_\infty \leq 1$ for all $\theta$ and $\mathbf{x} \in \mathcal{X}$.

Denote by $\mathbf{0}$ the constant function, mapping all inputs to 0. Let $\theta_t^\star$ be the following trajectory of SGD that is independent of the target. Define $\theta_t^\star$ recursively s.t. $\theta_0^\star = \theta_0$ and

$$\theta_{t+1}^\star = \theta_t^\star - \eta_t \nabla_\theta \left( L_\mathcal{D}(h_{\theta_t^\star}, \mathbf{0}) + R(\theta_t^\star) \right)$$

**Assumption 2** (Bounded error for gradient estimator). *For all $t$, suppose that*

$$\left\| \nabla L_{\mathcal{S}_t}(h_{\theta_t^\star}, \mathbf{0}) - \nabla L_\mathcal{D}(h_{\theta_t^\star}, \mathbf{0}) \right\|_\infty \leq \tau/2.$$

*Remark:* If $m = \tilde{O}(1/\tau^2)$,[2] Assumption 2 is satisfied w.h.p. for: 1) $\mathcal{S}_t \sim \mathcal{D}^m$ ("online" SGD), 2) $\mathcal{S}_t = \mathcal{S}$ for some $\mathcal{S} \sim \mathcal{D}^m$ ("offline" GD) and 3) $\mathcal{S}_t$ is a batch of size $m$ sampled uniformly at random from $\mathcal{S}$, where $\mathcal{S} \sim \mathcal{D}^M$ and $M \geq m$ ("offline" SGD). Indeed, in all these cases we have $\mathbb{E}_{\mathcal{S}_t} \left[ \nabla L_{\mathcal{S}_t}(h_{\theta_t^\star}, \mathbf{0}) \right] = \nabla L_\mathcal{D}(h_{\theta_t^\star}, \mathbf{0})$, and the above follows from standard concentration bounds.

The lower bound in this section uses standard statistical query arguments to show that gradient-based algorithms, *without sufficient resources*, will fail to learn $(n, k)$-parities. We therefore start by stating the four types of resources that impact learnability:

- The *number of parameters* $r$ – equivalently, the number of parallel "queries".
- The *number of gradient updates* $T$ – i.e. the serial running time of the training algorithm.
- The *gradient precision* $\tau$ – i.e. how close the empirical gradient is to the population gradient. As discussed above, $\tilde{O}(1/\tau^2)$ samples suffice to obtain such an estimator.
- The probability of success $\delta$.

The following theorem ties these four resources together, showing that without a sufficient allocation of these resources, gradient descent will fail to learn:

**Proposition 3.** *Assume that $\theta_0$ is randomly drawn from some distribution. For every $r, T, \delta, \tau > 0$, if $\frac{rT}{\tau^2 \delta} \leq \frac{1}{2} \binom{n}{k}$, then there exists some $(n, k)$-parity s.t. with probability at least $1 - \delta$ over the choice of $\theta_0$, the first $T$ iterates of SGD are statistically independent of the target function.*

The proof follows standard SQ lower bound arguments (Kearns, 1998; Feldman, 2008), and for completeness is given in Appendix B. The core idea of the proof is the observation that any gradient step has very little correlation (roughly $n^{-k}$) with most sparse-parity functions, and so there exists a parity function that has small correlation with all steps. In this case, the noise can force the gradient iterates to follow the trajectory $\theta_1^\star, \ldots, \theta_T^\star$, which is independent of the true target. We note that, while the focus of this paper is on sparse parities, similar analysis applies for a broader class of functions that are characterized by large SQ dimension (see Blum et al. (1994)).

Observe that there are various ways for an algorithm to escape the lower bound of Theorem 3. We can scale a single resource with $\binom{n}{k}$, keeping the rest small, e.g. by training a network of size $n^k$, or using a sample size of size $n^k$. Crucially, we note the possibility of *interpolating* between these extremes: one can spread this homogenized "cost" across multiple resources, by (e.g.) training a network of size $n^{k/2}$ using $n^{k/2}$ samples. In the next section, we show how neural networks can be tailored to solve the task in these interpolated resource scaling regimes.

---

[2] We use the notation $\tilde{O}$ to hide logarithmic factors.

## 3.2 Upper bounds: many ways to trade off the terms

As a warmup, we discuss some simple SQ algorithms that succeed in learning parities by properly scaling the different resources. First, consider deterministic exhaustive search, which computes the error of all possible $(n, k)$-parities, choosing the one with smallest error. This can be done with constant $\tau$ (thus, sample complexity logarithmic in $n$), but takes $T = \Theta(n^k)$ time. Since querying different parities can be done in parallel, we can reduce the number of steps $T$ by increasing the number of parallel queries $r$. Alternatively, it is possible to query only a randomly selected subset of parities, which reduces the overall number of queries but increases the probability of failure.

The above algorithms give a rough understanding of the frontier of algorithms that succeed at learning parities. However, at first glance, they do not seem to reflect algorithms used in deep learning, and are specialized to the parity problem. In this section, we will explore the ability of neural networks to achieve similar tradeoffs between the different resources. In particular, we focus on the interaction between sample complexity and network size, establishing learning guarantees with *interpolatable mixtures* of these resources.

Before introducing our main positive theoretical results, we discuss some prior theoretical results on learning with neural networks, and their limitations in the context of learning parities. Positive results on learning with neural networks can generally be classified into two categories: those that reduce the problem to convex learning of linear predictors over a predefined set of features (e.g. the NTK), and those that involve neural networks departing from the kernel regime by modifying the fixed features of the initialization, known as the feature learning regime.

**Kernel regime.** When neural networks trained with gradient descent stay close to their initial weights, optimization behaves like kernel regression on the neural tangent kernel (Jacot et al., 2018; Du et al., 2018): the resulting function is approximately of the form $\mathbf{x} \mapsto \langle \psi(\mathbf{x}), \mathbf{w} \rangle$, where $\psi$ is a *data-independent* infinite-dimensional embedding of the input, and $\mathbf{w}$ is some weighting of the features. However, it has been shown that the NTK (more generally, any fixed kernel) cannot achieve low $\ell_2$ error on the $(n, k)$-parity problem, unless the sample complexity grows as $\Omega(n^k)$ (see Kamath et al. (2020)). Thus, no matter how we scale the network size or training time, neural networks trained in the NTK regime cannot learn parities with low sample complexity, and thus do not enjoy the flexibility of resource allocation discussed above.

**Feature learning regime.** Due to the limitation of neural networks trained in the kernel regime, some works study learning in the "rich" regime, quantifying how hidden-layer features adapt to the data. Among these, Barak et al. (2022) analyze a feature learning mechanism requiring exceptionally small network width: SGD on 2-layer MLPs can solve the *online* sparse parity learning problem with network width *independent* of $n$ (dependent only on $k$), at the expense of requiring a suboptimal ($\approx n^k$) number of examples. This mechanism is *Fourier gap amplification*, by which SGD through a *single neuron* $\sigma(w^\top x)$ can perform feature selection in this setting, via exploiting a small gap between the relevant and irrelevant coordinates in the population gradient. The proof of Theorem 4 below relies on a similar analysis, extended to the offline regime (i.e., multiple passes over a dataset of limited size).

### 3.2.1 "Data × model size" success frontier for sparsely-initialized MLPs

In this section, we analyze a 2-layer MLP with ReLU ($\sigma(x) = \max(0, x)$) activation, trained with batch ("offline") gradient-descent over a sample $\mathcal{S}$ with $\ell_2$-regularized updates[3]:

$$\theta^{(t+1)} = (1 - \lambda^{(t)})\theta^{(t)} - \eta^{(t)}\nabla L_{\mathcal{S}}(h_{\theta^{(t)}})$$

We allow learning rates $\eta$, and weight decay coefficients $\lambda$ to differ between layers and iterations. For simplicity, we analyze the case where no additional noise is added to each update; however, we believe that similar results can be obtained in the noisy case (e.g., using the techniques in Feldman et al. (2017)). Finally, we focus on ReLU networks with $s$-sparse initialization of the first layer: every weight $\mathbf{w}_i$ has $s$ randomly chosen coordinates set to 1, and the rest set to 0. Note that after initialization, all of the network's weights are allowed to move, so sparsity is not necessarily preserved during training.

---

[3]For simplicity, we do not assume adversarial noise in the gradients as in the lower bound. However, similar results can be shown under bounded noise.

**Over-sparse initialization ($s > \Omega(k)$).** The following theorem demonstrates a "data $\times$ width" success frontier when learning $(n, k)$-parities with sparsely initialized ReLU MLPs at sparsity levels $s > \Omega(k)$.

**Theorem 4.** *Let $k$ be an even integer, and $\epsilon \in (0, 1/2)$. Assume that $n \geq \Omega(1/\epsilon^2)$. For constants $c_1, c_2, c_3, c_4$ depending only on $k$, choose the following: (1) sparsity level: $s \geq c_1/\epsilon^2$, for some odd $s$, (2) width of the network: $r = c_2(n/s)^k$, (3) sample size: $m \geq c_3(s/k)^{k-1}n^2 \log n$, and (4) number of iterations: $T \geq c_4/\epsilon^2$. Then, for every $(n, k)$-parity distribution $\mathcal{D}$, with probability at least $0.99$ over the random samples and initialization, gradient descent with these parameter settings returns a function $h_T$ s.t. $L_{\mathcal{D}}(h_T) \leq \epsilon$.*

Intuitively, by varying the sparsity parameter in Theorem 4, we obtain a family of algorithms which smoothly interpolate between the *small-data/large-width* and *large-data/small-width* regimes of tractability. First, consider a sparsity level linear in $n$ (i.e. $s = \alpha \cdot n$). In this case, a small network (with width $r$ independent of the input dimension) is sufficient for solving the problem, but the sample size must be large ($\Omega(n^{k+1})$) for successful learning; this recovers the result of Barak et al. (2022). At the other extreme, if the sparsity is independent of $n$, the sample complexity grows only as $O(n^2 \log n)^4$, but the requisite width becomes $\Omega(n^k)$.

*Proof sketch.* The proof of Theorem 4 relies on establishing a Fourier anti-concentration condition, separating the relevant (i.e. indices in $S$) and irrelevant weights in the initial population gradient, similarly as the main result in Barak et al. (2022). When we initialize an $s$-sparse neuron, there is a probability of $\gtrsim (s/n)^k$ that the subset of activated weights contains the "correct" subset $S$. In this case, to detect the subset $S$ via the Fourier gap, it is sufficient to observe $s^{k-1}$ examples instead of $n^{k-1}$. Initializing the neurons more sparsely makes it less probable to draw a *lucky* neuron, but once we draw a lucky neuron, it requires fewer samples to find the right features. Thus, increasing the width reduces overall sample complexity, by sampling a large number of "lottery tickets".

**Under-sparse initialization ($s < k$).** The sparsity parameter can modulate similar data vs. width tradeoffs for feature learning in the "under-sparse" regime. We provide a partial analysis for this more challenging case, showing that one step of gradient descent can recover a correct subnetwork. Appendix B.3 discusses the mathematical obstructions towards obtaining an end-to-end guarantee of global convergence.

**Theorem 5.** *For even $k, s$, sparsity level $s < k$, network width $r = O((n/k)^s)$, and $\varepsilon$-perturbed[5] $s$-sparse random initialization scheme s.t. for every $(n, k)$-parity distribution $\mathcal{D}$, with probability at least $0.99$ over the choice of sample and initialization after one step of batch gradient descent (with gradient clipping) with sample size $m = O((n/k)^{k-s-1})$ and appropriate learning rate, there is a subnetwork in the ReLU MLP that approximately computes the parity function.*

Here, the sample complexity can be improved by a factor of $n^s$, at the cost of requiring the width to be $n^s$ times larger. The proof of Theorem 5 relies on a novel analysis of improved Fourier gaps with "partial progress": intuitively, if a neuron is randomly initialized with a subset of the relevant indices $S$, it only needs to identify $k - s$ more coordinates, inheriting the improved sample complexity for the $(n - s, k - s)$-parity problem. Note that the probability of finding such a lucky neuron scales as $(n/k)^{-s}$, which governs how wide (number of lottery tickets) the network needs to be.

**Remarks on the exact-sparsity regime.** Our theoretical analyses do not extend straightforwardly to the case of $s = \Theta(k)$. Observe that if the value of $k$ is known, initializing a network of size $r = \widetilde{O}\left(n^k\right)$ with sparsity $s = k$ gives w.h.p. a subnetwork with good first-layer features at initialization. We believe that with a proper choice of regularization and training scheme, it is possible to show that such a network learns to select this subnetwork with low sample complexity. We leave the exact details of this construction and end-to-end proofs for future work.

**Analogous results for dense initialization schemes?** We believe that the principle of "parallel search with randomized per-subnetwork sample complexities" extends to other initialization schemes

---

[4]We note that the additional $n^2$ factor in the sample complexity can be removed if we apply gradient truncation, thus allowing only a logarithmic dependence on $n$ in the small-sample case.

[5]For ease of analysis, we use a close variant of the sparse initialization scheme: $s$ coordinates out of the $n$ coordinates are chosen randomly and set to 1, and the rest of the coordinates are set to $\epsilon < (n - s)^{-1}$. Without the small norm dense component in the initialization, the population gradient will be 0 at initialization.

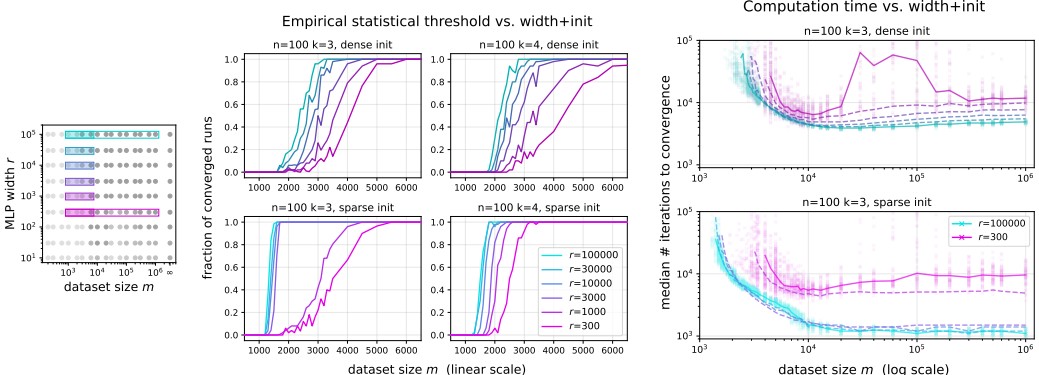

Figure 2: Zoomed-in views of the interactions between width, data, time, and luck. *Left:* Locations of these runs in the larger parameter space. *Center:* Success probability vs. dataset size. In accordance with our theory, **width buys luck, and improves end-to-end sample efficiency**, despite increasing the network's capacity. *Right:* Number of iterations to convergence vs. dataset size. We observe a **data vs. time tradeoff** in the grokking regime (at the edge of feasibility), as well as a **"sample-wise double descent"** performance drop with more data (which can also be seen in Figure 1, and disappears with larger widths). Comparing dense vs. sparse initializations (upper vs. lower plots; sparse inits are colored more brightly), we see computational and statistical benefits of the sparse initialization scheme from Section 3.2.1.

(including those more commonly used in practice), and leads to analogous success frontiers. To support this, our experiments investigate both sparse and uniform initialization, with qualitatively similar findings. For dense initializations, the mathematical challenge lies in analyzing the Fourier anti-concentration of general halfspaces (see the discussion and experiments in Appendix C.1 of (Barak et al., 2022)). The axis-aligned inductive biases imparted by sparse initialization may also be of independent practical interest.

# 4 Experiments

A high-level takeaway from Section 3 is that when a learning problem is computationally difficult but statistically easy, a complex frontier of resource tradeoffs can emerge; moreover, it is possible to interpolate between extremes along this frontier using ubiquitous algorithmic choices in deep learning, such as overparameterization, random initialization, and weight decay. In this section, we explore the nature of the frontier with an empirical lens—first with end-to-end sparse parity learning, then with natural tabular datasets.

## 4.1 Empirical Pareto frontiers for offline sparse parity learning

We launch a large-scale ($\sim$200K GPU training runs) exploration of resource tradeoffs when training neural networks to solve the offline sparse parity problem. While Section 3 analyzes idealized variants of SGD on MLPs which interpolate along the problem's resource tradeoff frontier, in this section we ask whether the same can be observed end-to-end with standard training and regularization.

On various instances of the $(n, k)$-sparse parity learning problem, we train a 2-layer MLP with identical hyperparameters, varying the network width $r \in \{10, 30, 100, \dots, 10^5\}$ and the dataset size $m \in \{100, 200, 300, 600, 1000, \dots, 10^6\}$. Alongside standard algorithmic choices, we consider one non-standard augmentation of SGD: the *under-sparse* initialization scheme from Section 3.2.1; we have proven that these give rise to "lottery ticket" neurons which learn the influential coordinates more sample-efficiently. Figure 1 (in the introduction) and Figure 2 illustrate our findings at a high level; details and additional discussion are in Appendix C.1). We list our key findings below:

(1) **A "success frontier": large width can compensate for small datasets.** We observe convergence and perfect generalization when $m \ll n^k$. In such regimes, which are far outside the

online setting considered by Barak et al. (2022), high-probability sample-efficient learning is enabled by large width. This can be seen in Figure 1 (left), and analogous plots in Appendix C.1.

(2) **Width is monotonically beneficial, and buys data, time, and luck.** In this setting, increasing the model size yields exclusively positive effects on success probability, sample efficiency, and the number of serial steps to convergence (see Figure 2). This is a striking example where end-to-end generalization behavior runs *opposite* to uniform convergence-based upper bounds, which predict that enlarging the model's capacity *worsens* generalization.

(3) **Sparse axis-aligned initialization buys data, time, and luck.** Used in conjunction with a wide network, we observe that a sparse, axis-aligned initialization scheme yields strong improvements on all of these axes; see Figure 2 (bottom row). In smaller hyperparameter sweeps, we find that $s = 2$ (i.e. initialize every hidden-layer neuron with a random 2-hot weight vector) works best.

(4) **Intriguing effects of dataset size.** As we vary the sample size $m$, we note two interesting phenomena; see Figure 2 (right). The first is grokking (Power et al., 2022), which has been previously documented in this setting (Barak et al., 2022; Merrill et al., 2023). This entails a *data vs. time* tradeoff: for small $m$ where learning is marginally feasible, optimization requires significantly more training iterations $T$. Our second observation is a "sample-wise double descent" (Nakkiran et al., 2021): success probability and convergence times can worsen with *increasing* data. Both of these effects are also evident in Figure 1).

**Lottery ticket neurons.** The above findings are consistent with the viewpoint taken by the theoretical analysis, where randomly-initialized SGD plays the role of *parallel search*, and a large width increases the number of random subnetworks available for this process—in particular, the "winning lottery ticket" neurons, for which feature learning occurs more sample-efficiently. To provide further evidence that sparse subnetworks are responsible for learning the parities, we perform a smaller-scale study of network prunability in Appendix C.2.

## 4.2 Sample-efficient deep learning on natural tabular datasets

Sparse parity learning is a toy problem, in that it is defined by an idealized distribution, averting the ambiguities inherent in reasoning about real-life datasets. However, due to its provable hardness (Theorem 3, as well as the discussion in Section 2.1), it is a *maximally hard* toy problem in a rigorous sense[6]. In this section, we perform a preliminary investigation of how the empirical and algorithmic insights gleaned from Section 4.1 can be transferred to more realistic learning scenarios.

To this end, we use the benchmark assembled by Grinsztajn et al. (2022), a work which specifically investigates the performance gap between neural networks and tree-based classifiers (e.g. random forests, gradient-boosted trees), and includes a standardized suite of 16 classification benchmarks with numerical input features. The authors identify three common aspects of tabular[7] tasks which present difficulties for neural networks, especially vanilla MLPs:

(i) **The feature spaces are not rotationally invariant.** In state-of-the-art deep learning, MLPs are often tasked with function representation in rotation-invariant domains (token embedding spaces, convolutional channels, etc.).

(ii) **Many of the features are uninformative.** In order to generalize effectively, especially from a limited amount of data, it is essential to avoid overfitting to these features.

(iii) **There are meaningful high-frequency/non-smooth patterns in the target function.** Combined with property (i), decision tree-based methods (which typically split on axis-aligned features) can appear to have the ideal inductive bias for tabular modalities of data.

Noting that the sparse parity task possesses all three of the above qualities, we conduct a preliminary investigation on whether our empirical findings in the synthetic case carry over to natural tabular data. In order to study the impact of algorithmic choices (mainly width and sparse initialization) on sample efficiency, we create low-data problem instances by subsampling varying fractions of each dataset for training. Figure 3 provides a selection of our results. We note the following empirical findings, which are the tabular data counterparts of results (2) and (3) in Section 4.1:

---

[6]Namely, its SQ dimension is equal to the number of hypotheses, which is what leads to Theorem 3.

[7]"Tabular data" refers to the catch-all term for data sources where each coordinate has a distinct semantic meaning which is consistent across points.

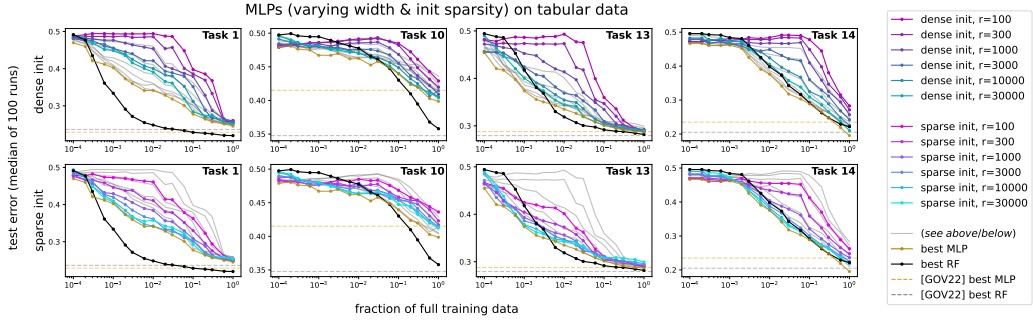

Figure 3: Analogous investigation of MLP width $r$ and sparse initialization for real-world tabular datasets (OpenML benchmarks assembled by Grinsztajn et al. (2022)), varying the dataset size $m$ via downsampling. Large width and sparse initialization tend to improve generalization, in accordance with the theory and experiments for synthetic parity tasks. In some settings, our best MLPs outperform tuned random forests. Dotted lines denote the test errors reported by Grinsztajn et al. (2022) of tuned MLPs and RFs on the full datasets. Full results on all 16 tasks are in Appendix C.3.

(2T) **Wide networks generalize on small tabular datasets.** Like in the synthetic experiments, width yields nearly monotonic end-to-end benefits for learning. This suggests that the "parallel search + pruning" mechanisms analyzed in our paper are also at play in these settings. In some (but not all) cases, these MLPs perform competitively with tuned tree-based classifiers.

(3T) **Sparse axis-aligned initialization sometimes improves end-to-end performance.** This effect is especially pronounced on datasets which are downsampled to be orders of magnitude smaller. We believe that this class of drop-in replacements for standard initialization merits further investigation, and may contribute to closing the remaining performance gap between deep learning and tree ensembles on small tabular datasets.

## 5  Conclusion

We have presented a theoretical and empirical study of offline sparse parity learning with neural networks; this is a provably hard problem which admits a multi-resource lower bound in the SQ model. We have shown that the lower bound can be surmounted using varied mixtures of these resources, which correspond to natural algorithmic choices and scaling axes in deep learning. By investigating how these choices influence the empirical "success frontier" for this hard synthetic problem, we have arrived at some promising improvements for MLP models of tabular data (namely, large width and sparse initialization). These preliminary experiments suggest that a more intensive, exhaustive study of algorithmic improvements for MLPs on tabular data has a chance of reaping significant rewards, perhaps even surpassing the performance of decision tree ensembles.

**Broader impacts and limitations.**    The nature of this work is foundational; the aim of our theoretical and empirical investigations is to contribute to the fundamental understanding of neural feature learning, and the influences of scaling relevant resources. A key limitation is that our benchmarks on tabular data are only preliminary; it is a significant and perennial methodological challenge to devise fair and comprehensive comparisons between neural networks and tree-based learning paradigms.

## Acknowledgements

We are grateful to Boaz Barak for helpful discussions and to Matthew Salganik for helpful comments on a draft version. This work has been made possible in part by a gift from the Chan Zuckerberg Initiative Foundation to establish the Kempner Institute for the Study of Natural and Artificial Intelligence. Sham Kakade acknowledges funding from the Office of Naval Research under award N00014-22-1-2377. Ben Edelman acknowledges funding from the National Science Foundation Graduate Research Fellowship Program under award #DGE 2140743.

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
