# Part I

# Appendix

## Table of Contents

# A    Additional related work

**Learning parities with neural networks.**    Parities (or XORs) have been shown to be computationally hard to learn for SQ algorithms including gradient-based methods. Various works make additional assumptions to avoid these hardness results to show that neural networks can be efficiently trained to learn parities (Daniely and Malach, 2020; Shi et al., 2021; Frei et al., 2022; Malach et al., 2021). More recently, Barak et al. (2022) have focused on understanding how neural networks training on parities without any additional assumption behaves at this computational-statistical limit. They show that one step of gradient descent on a single neuron is able to recover the indices corresponding to the parity with $n^{O(k)}$ samples/computation. Abbe et al. (2023) improve this bound to $O(n^{k-1})$ online SGD steps and generalize the result to handle hierarchical staircases of parity functions which requires a multi-step analysis. Telgarsky (2022) studies the problem of 2-sparse parities with two-layer neural networks trained with vanilla SGD (unlike our restricted two-step training algorithm) and studies the margins achieved post training. They use the margins to get optimal sample complexity $\tilde{O}(n^2/\epsilon)$ in the NTK regime. Going beyond NTK, they analyze gradient flow (with certain additional modifications) on an exponential wide 2-layer network (making it computationally inefficient) to get the improved sample complexity of $\tilde{O}(n/\epsilon)$. In contrast to this, our goal is to improve sample complexity while maintaining computational efficiency, using random guessing via the sparse initialization.

**Learning single-index/multi-index models over Gaussians with neural networks.**    Another line of work (Ben Arous et al., 2021; Ba et al., 2022; Damian et al., 2022; Bietti et al., 2022; Damian et al., 2023) has focused on learning functions that depend on a few directions, in particular, single-index and multi-index models over Gaussians using neural nets. These can be thought of as a continuous analog to our sparse parity problem. In a similar analysis (as parities) of online SGD for single index models, Ben Arous et al. (2021) propose the notion of an *information exponent* which captures the initial correlation between the model and the target function, and get convergence results similar to the parity setting with sample complexity $O(n^{k-1})$ for information exponent $k$ (can be thought similar to the $k$ in the parity learning problem). Damian et al. (2023) improve this result by showing that a smoothed version of GD achieves the optimal sample complexity (for CSQ algorithms) of $(n^{k/2})$. Going beyond CSQ algorithms, Chen and Meka (2020) provide a filtered-PCA algorithm that achieves polynomial dependence on the dimension $n$ in both compute and sample complexity. Note that this is not achievable for CSQ algorithms. For the parity learning problem, the SQ computational lower bounds are $\Omega(n^k)$ (as described in Section 3.1).

**Empirical inductive biases of large MLPs.**    Our experiments on tabular benchmarks suggest that wide and sparsely-initialized vanilla MLPs can sometimes close the performance gap between neural networks and decision tree ensemble methods. This corroborates recent findings that vanilla MLPs have strong enough inductive biases to generalize nontrivially in natural data modalities, despite the overparameterization and lack of architectural biases via convolution or recurrence. Notably, many state-of-the-art computer vision models have removed convolutions (Dosovitskiy et al., 2020; Tolstikhin et al., 2021); recently, (Bachmann et al., 2023) demonstrate that even large vanilla NLPs can compete with convolutional models for image classification. Yang et al. (2022a) find monotonic improvements in terms of model width, which are stabilized by their theoretically-motivated hyperparameter scaling rules.

**Multi-resource scaling laws for deep learning.**    Many empirical studies (Kaplan et al., 2020; Henighan et al., 2020; Hoffmann et al., 2022; Zhai et al., 2022), motivated by the pressing need to allocate resources effectively in large-scale deep learning, corroborate the presence and regularity of neural scaling laws. Precise statements and hypotheses vary; Kaplan et al. (2020) fit power-law expressions which predict holdout validation log-perplexity of a language model in terms of dataset size, model size, and training iterations ($m, r, T$ in our notation). The present work shows how such a joint dependence on $m \times r \times T$ can arise from a single feature learning problem with a computational-statistical gap. Numerous works attempt to demystify neural scaling laws with theoretical models (Bahri et al., 2021; Hutter, 2021; Michaud et al., 2023); ours is unique in that it does not suppose a long-tailed data distribution (the statistical complexity of identifying a sparse parity is benign). We view these accounts to be mutually compatible: we *do not* purport that statistical query complexity is the unique origin of neural scaling laws, nor that there is a *single* such mechanism.

# B  Proofs

## B.1  Multi-resource lower bound for sparse parity learning

For some target function $f$ and some parameters $\theta$, we denote the population gradient over the distribution $\mathcal{D}$ by:

$$g(f, \theta) = \mathbb{E}_{\mathbf{x} \sim \mathcal{D}} [\nabla_\theta \ell(h_\theta(\mathbf{x}), f(\mathbf{x}))]$$

and we denote by $g_i(\cdot, \cdot)$ the gradient w.r.t. the $i$-th coordinate of $\theta$.

Similarly, denote the empirical gradient by:

$$\hat{g}(f, \theta) = \frac{1}{m} \sum_{\mathbf{x} \in \mathcal{S}} \nabla_\theta \ell(h_\theta(\mathbf{x}), f(\mathbf{x}))$$

and $\hat{g}_i(\cdot, \cdot)$ denotes the $i$-th coordinate of the empirical gradient.

**Lemma 6.** *For every $\theta$ and every $i$ it holds that*

$$\mathbb{E}_{S \sim \binom{n}{k}} \left[ \left( g_i(\chi_S, \theta) - \ell_0(h_\theta(\mathbf{x})) \cdot \frac{\partial}{\partial \theta_i} h_\theta(\mathbf{x}) \right)^2 \right] \leq \frac{1}{\binom{n}{k}}$$

*Proof.* Fix some $i \in [r]$,

$$\mathbb{E}_{S \sim \binom{n}{k}} \left[ \left( g_i(\chi_S, \theta) - \ell_0(h_\theta(\mathbf{x})) \cdot \frac{\partial}{\partial \theta_i} h_\theta(\mathbf{x}) \right)^2 \right]$$

$$= \mathbb{E}_{S \sim \binom{n}{k}} \left[ \mathbb{E}_{\mathbf{x} \sim \mathcal{D}} \left[ \chi_S(\mathbf{x}) \cdot \frac{\partial}{\partial \theta_i} h_\theta(\mathbf{x}) \right]^2 \right]$$

$$= \frac{1}{\binom{n}{k}} \sum_{S \in \binom{n}{k}} \mathbb{E}_{\mathbf{x} \sim \mathcal{D}} \left[ \chi_S(\mathbf{x}) \cdot \frac{\partial}{\partial \theta_i} h_\theta(\mathbf{x}) \right]^2 \leq \frac{1}{\binom{n}{k}}$$

where the last inequality is from Parseval, using the assumption $\|\nabla h_\theta(\mathbf{x})\|_\infty \leq 1$.

$\square$

*Proof of Proposition 3.* Using the Lemma 6 we get:

$$\mathbb{E}_{S \sim \binom{n}{k}} \left[ \max_{i,t} \left( g_i(\chi_S, \theta_t^*) - \ell_0(h_{\theta_t^*}(\mathbf{x})) \cdot \frac{\partial}{\partial \theta_i} h_{\theta_t^*}(\mathbf{x}) \right)^2 \right]$$

$$\leq \mathbb{E}_{S \sim \binom{n}{k}} \left[ \sum_{i=1}^r \sum_{t=1}^T \left( g_i(\chi_S, \theta_t^*) - \ell_0(h_{\theta_t^*}(\mathbf{x})) \cdot \frac{\partial}{\partial \theta_i} h_{\theta_t^*}(\mathbf{x}) \right)^2 \right]$$

$$\leq \frac{rT}{\binom{n}{k}} \leq \delta\tau^2$$

Therefore, taking expectation over the choice of $\theta_0$

$$\mathbb{E}_{\theta_0} \mathbb{E}_{S \sim \binom{n}{k}} \left[ \max_{i,t} \left( g_i(\chi_S, \theta_t^*) - \ell_0(h_{\theta_t^*}(\mathbf{x})) \cdot \frac{\partial}{\partial \theta_i} h_{\theta_t^*}(\mathbf{x}) \right)^2 \right]$$

$$= \mathbb{E}_{S \sim \binom{n}{k}} \mathbb{E}_{\theta_0} \left[ \max_{i,t} \left( g_i(\chi_S, \theta_t^*) - \ell_0(h_{\theta_t^*}(\mathbf{x})) \cdot \frac{\partial}{\partial \theta_i} h_{\theta_t^*}(\mathbf{x}) \right)^2 \right] \leq \delta\tau^2/2$$

So, there exists some $S \in \binom{n}{k}$ s.t.

$$\mathbb{E}_{\theta_0} \left[ \max_{i,t} \left( g_i(\chi_S, \theta_t^*) - \ell_0(h_{\theta_t^*}(\mathbf{x})) \cdot \frac{\partial}{\partial \theta_i} h_{\theta_t^*}(\mathbf{x}) \right)^2 \right] \leq \delta\tau^2/2$$

Observe the noise variable $\xi_t = \ell_0(h_{\theta_t^*}(\mathbf{x})) \cdot \frac{\partial}{\partial \theta_i} h_{\theta_t^*}(\mathbf{x}) - \hat{g}_i(\chi_S, \theta_t^*, \mathcal{S}_t)$. From Markov's inequality and Assumption 2, with probability at least $1 - \delta$ over the choice of $\theta_0$, for all $t \leq T$ and $i \in [r]$:

$$\left| \hat{g}_i(\chi_S, \theta_t^*, \mathcal{S}_t) - \ell_0(h_{\theta_t^*}(\mathbf{x})) \cdot \frac{\partial}{\partial \theta_i} h_{\theta_t^*}(\mathbf{x}) \right| \leq \tau$$

therefore, we get that $\xi_1, \ldots, \xi_T \in [-\tau, \tau]^r$ (i.e., this is a valid choice of adversarial noise variables), and SGD follows the trajectory $\theta_1^\star, \ldots, \theta_T^\star$. $\qquad\square$

## B.2 Feature selection with an over-sparse initialization and a wide network

### B.2.1 Warmup: existence of good subnetworks

Let $h_{\mathbf{w}}(\mathbf{x}) = \sigma(\langle \mathbf{w}, \mathbf{x} \rangle)$ be a single ReLU neuron, where $\sigma(x) = \max\{x, 0\}$. Fix some $4k < s \leq n$. Assume we initialize $\mathbf{w} \in \{0, 1\}^n$ by randomly choosing $s$ coordinates and setting them to 1, and setting the rest to zero. Fix some subset $S \subseteq \binom{n}{k}$. We say that $\mathbf{w}$ is a *good* neuron if $S \subseteq \mathbf{w}$. We say that $\mathbf{w}$ is a *bad* neuron if it is not a *good* neuron.

**Lemma 7.** *With probability at least* $(s/2n)^k$ *over the choice of* $\mathbf{w}$, $\mathbf{w}$ *is a* good *neuron.*

*Proof.* There are $\binom{n}{s}$ choices for $\mathbf{w}$, and there are $\binom{n-k}{s-k}$ *good* choices for $\mathbf{w}$. Observe that:

$$\binom{n}{s} = \frac{n(n-1)\cdots(n-k+1)}{s(s-1)\cdots(s-k+1)} \binom{n-k}{s-k} \leq \left(\frac{2n}{s}\right)^k \binom{n-k}{s-k}$$

and therefore the required follows. $\qquad\square$

**Lemma 8.** *Assume $k$ is even, $s$ is odd and $k < \frac{s}{4}$. There exist constants $C_k, c_k$ s.t. if $\mathbf{w}$ is a* good *neuron, then:*

1. *For all $i \in S$,*

$$\mathbb{E}_{\mathbf{x}}\left[ \frac{\partial}{\partial w_i} h_{\mathbf{w}}(\mathbf{x}) \cdot \chi_S(\mathbf{x}) \right] = C_{k,s}$$

2. *For all $i \notin S$,*

$$\mathbb{E}_{\mathbf{x}}\left[ \frac{\partial}{\partial w_i} h_{\mathbf{w}}(\mathbf{x}) \cdot \chi_S(\mathbf{x}) \right] = w_i c_{k,s}$$

*and furthermore, there exists a constant $\kappa_k$ s.t. $|C_{k,s}| > \kappa_k \binom{s}{k-1}^{-1/2}$ and $\frac{|c_{k,s}|}{|C_{k,s}|} \leq \frac{4k}{s}$.*

*Proof.* First, consider the case where $i \in S$. Therefore,

$$
\begin{aligned}
\mathbb{E}_{\mathbf{x} \sim \{\pm 1\}^n}\left[ \frac{\partial}{\partial w_i} h_{\mathbf{w}}(\mathbf{x}) \cdot \chi_S(\mathbf{x}) \right] &= \mathbb{E}_{\mathbf{x} \sim \{\pm 1\}^n}\left[ \sigma'(\langle \mathbf{w}, \mathbf{x} \rangle) \cdot x_i \cdot \chi_S(\mathbf{x}) \right] \\
&= \mathbb{E}_{\mathbf{x} \sim \{\pm 1\}^n}\left[ \mathbf{1}_{\{\sum_{j \in \mathbf{w}} x_j \geq 0\}} \chi_{S \setminus \{i\}}(\mathbf{x}) \right] \\
&= \mathbb{E}_{\mathbf{x} \sim \{\pm 1\}^s}\left[ \left( \frac{1}{2} \mathrm{Maj}_s(\mathbf{x}) + \frac{1}{2} \right) \chi_{S \setminus \{i\}}(\mathbf{x}) \right] \\
&= \frac{1}{2} \widehat{\mathrm{Maj}_s}(S \setminus \{i\})
\end{aligned}
$$

where $S \setminus \{i\}$ is interpreted as a subset of $[s]$. From symmetry of the Majority function, all Fourier coefficients of the same order are equal. Therefore, the first condition holds for $C_{k,s} = \frac{1}{2}\widehat{\mathrm{Maj}_s}(k-1)$, where $\widehat{\mathrm{Maj}_s}(k-1)$ denotes the $(k-1)$-th order Fourier coefficient.

When $i \in S \setminus \mathbf{w}$ we get:

$$\mathop{\mathbb{E}}_{\mathbf{x} \sim \{\pm 1\}^n} \left[ \frac{\partial}{\partial w_i} h_{\mathbf{w}}(\mathbf{x}) \cdot \chi_S(\mathbf{x}) \right] = \mathop{\mathbb{E}}_{\mathbf{x} \sim \{\pm 1\}^n} \left[ \sigma'(\langle \mathbf{w}, \mathbf{x} \rangle) \cdot x_i \cdot \chi_S(\mathbf{x}) \right]$$

$$= \mathop{\mathbb{E}}_{\mathbf{x} \sim \{\pm 1\}^n} \left[ \mathbf{1}_{\{\sum_{j \in \mathbf{w}} x_j > 0\}} \chi_{S \cup \{i\}}(\mathbf{x}) \right]$$

$$= \mathop{\mathbb{E}}_{\mathbf{x} \sim \{\pm 1\}^s} \left[ \left( \frac{1}{2} \mathrm{Maj}_s(\mathbf{x}) + \frac{1}{2} \right) \chi_{S \cup \{i\}}(\mathbf{x}) \right]$$

$$= \frac{1}{2} \widehat{\mathrm{Maj}_s}(S \cup \{i\})$$

Finally, when $i \notin S$ we get:

$$\mathop{\mathbb{E}}_{\mathbf{x} \sim \{\pm 1\}^n} \left[ \frac{\partial}{\partial w_i} h_{\mathbf{w}}(\mathbf{x}) \cdot \chi_S(\mathbf{x}) \right] = \mathop{\mathbb{E}}_{\mathbf{x} \sim \{\pm 1\}^n} \left[ \sigma'(\langle \mathbf{w}, \mathbf{x} \rangle) \cdot x_i \cdot \chi_S(\mathbf{x}) \right]$$

$$= \mathbb{E} \left[ \sigma'(\langle \mathbf{w}, \mathbf{x} \rangle) \cdot \chi_S(\mathbf{x}) \right] \mathop{\mathbb{E}}_{x_i} [x_i] = 0$$

Therefore, the second condition holds with $c_k = \frac{1}{2} \widehat{\mathrm{Maj}_s}(k + 1)$.

Now, from Theorem 5.22 in O'Donnell (2014) we have:

$$\binom{s}{k-1} \cdot \widehat{\mathrm{Maj}_s}(k-1)^2 = \sum_{S' \in \binom{s}{k-1}} \widehat{\mathrm{Maj}_s}(S')^2 \geq \rho(k-1)$$

where $\rho(v) = \frac{2}{\pi v 2^v} \binom{v-1}{\frac{v-1}{2}}$. So, we get $|C_{k,s}| \geq \frac{1}{2} \sqrt{\rho(k-1)} \binom{s}{k-1}^{-1/2}$. Using the same Theorem, we also have:

$$\binom{s}{k+1} \cdot \widehat{\mathrm{Maj}_s}(k+1)^2 = \sum_{S' \in \binom{s}{k+1}} \widehat{\mathrm{Maj}_s}(S')^2 \leq 2\rho(k+1) < 2\rho(k-1)$$

So, we get:

$$\frac{|c_{k,s}|}{|C_{k,s}|} \leq \frac{\sqrt{2\rho(k-1)} \binom{s}{k+1}^{-1/2}}{\sqrt{\rho(k-1)} \binom{s}{k-1}^{-1/2}} = \sqrt{2} \sqrt{\frac{\binom{s}{k+1}}{\binom{s}{k-1}}}$$

$$= \sqrt{\frac{2k(k+1)}{(s-k+2)(s-k+1)}} \leq \sqrt{\frac{4k^2}{(1/4)s^2}} = 4\frac{k}{s}$$

$\square$

**Lemma 9.** *Assume that $k < \frac{s}{4}$. If $\mathbf{w}$ is a* bad *neuron, then:*

$$\left\| \mathop{\mathbb{E}}_{\mathbf{x}} \left[ \frac{\partial}{\partial \mathbf{w}} h_{\mathbf{w}}(\mathbf{x}) \cdot \chi_S(\mathbf{x}) \right] \right\|_1 \leq C_{k,s}$$

*Proof.* First, assume that $|S \setminus \mathbf{w}| \geq 2$. In this case, there exist $i, i' \in S$ s.t. $w_i = w_{i'} = 0$ and $i \neq i'$. Fix some $j \in [n]$, and choose some $j' \in i, i'$ s.t. $j' \neq j$.

$$\mathop{\mathbb{E}}_{\mathbf{x} \sim \{\pm 1\}^n} \left[ \frac{\partial}{\partial w_j} h_{\mathbf{w}}(\mathbf{x}) \cdot \chi_S(\mathbf{x}) \right] = \mathop{\mathbb{E}}_{\mathbf{x} \sim \{\pm 1\}^n} \left[ \sigma'(\langle \mathbf{w}, \mathbf{x} \rangle) \cdot \chi_S(\mathbf{x}) \cdot x_j \right]$$

$$= \mathop{\mathbb{E}}_{\mathbf{x} \sim \{\pm 1\}^n} \left[ \sigma'(\langle \mathbf{w}, \mathbf{x} \rangle) \cdot \chi_{S \setminus \{j'\}}(\mathbf{x}) \cdot x_j x_{j'} \right]$$

$$= \mathop{\mathbb{E}}_{x_j'} [x_{j'}] \cdot \mathop{\mathbb{E}}_{\mathbf{x}_{[n] \setminus \{j'\}}} \left[ \sigma'(\langle \mathbf{w}, \mathbf{x} \rangle) \cdot \chi_{S \setminus \{j'\}}(\mathbf{x}) \cdot x_j \right] = 0$$

and this gives the required.

Now, assume that $S \setminus \mathbf{w} = \{i\}$ for some index $i$. For every $j \neq i$, similarly to the previous analysis, we have:

$$\mathbb{E}_{\mathbf{x} \sim \{\pm 1\}^n} \left[ \frac{\partial}{\partial w_j} h_{\mathbf{w}}(\mathbf{x}) \cdot \chi_S(\mathbf{x}) \right] = 0$$

Finally, similarly to the proof of Lemma 8, we have:

$$\mathbb{E}_{\mathbf{x} \sim \{\pm 1\}^n} \left[ \frac{\partial}{\partial w_i} h_{\mathbf{w}}(\mathbf{x}) \cdot \chi_S(\mathbf{x}) \right] = \frac{1}{2} \widehat{\mathrm{Maj}_s}(S \setminus \{i\}) = C_{k,s}$$

and so we get the required. $\qquad \square$

### B.2.2 End-to-end result

We train the following network:

$$f_{\mathbf{W}, \mathbf{b}, \mathbf{u}, \beta}(\mathbf{x}) = \sum_{i=1}^r u_i \sigma \left( \langle \mathbf{w}_i, \mathbf{x} \rangle + b_i \right) + \beta$$

We fix some $k \leq s < n$ and initialize the network as follows:

- Randomly initialize $\mathbf{w}_1, \ldots, \mathbf{w}_{r/2} \in \{0, 1\}^n$ s.t. $\|\mathbf{w}_i\|_1 = s$ (i.e., each $\mathbf{w}_i$ has $s$ active coordinates), with a uniform distribution over all $\binom{n}{s}$ subsets.

- Randomly initialize $b_1, \ldots, b_{r/2} \sim \{\beta_1, \ldots, \beta_{k/2-1}\}$ where $\beta_i = \frac{1}{2k}(-k + 2i + 1/16)$.

- Randomly initialize $u_1, \ldots, u_{r/2} \sim \{\pm 1\}$ uniformly at random.

- Initialize $\mathbf{w}_{r/2+1}, \ldots, \mathbf{w}_r, b_{r/2+1}, \ldots, b_r$ s.t. $\mathbf{w}_i = \mathbf{w}_{i-r/2}$ and $b_i = b_{i-r/2}$ (symmetric initialization).

- Initialize $u_{r/2+1}, \ldots, u_r$ s.t. $u_i = -u_{i-r/2}$.

- Initialize $\beta = 0$

Let $\ell(\hat{y}, y) = \max(1 - y\hat{y}, 0)$ be the hinge-loss function. Given some distribution $\mathcal{D}$ over $\mathcal{X} \times \{\pm 1\}$, define the loss of $f$ over the distribution by:

$$L_{\mathcal{D}}(f) = \mathbb{E}_{(\mathbf{x}, y) \sim \mathcal{D}} [\ell(f(\mathbf{x}), y)]$$

Similarly, given a sample $\mathcal{S} \subseteq \mathcal{X} \times \{\pm 1\}$, define the loss of $f$ on the sample by:

$$L_{\mathcal{S}}(f) = \frac{1}{|\mathcal{S}|} \sum_{(\mathbf{x}, y) \in \mathcal{S}} \ell(f(\mathbf{x}), y)$$

We train the network by gradient descent on a sample $\mathcal{S}$ with $\ell_2$ regularization (weight decay):

$$\theta^{(t+1)} = (1 - \lambda^{(t)})\theta^{(t)} - \eta^{(t)} \nabla L_{\mathcal{S}}(f_{\theta^{(t)}})$$

We allow choosing learning rate $\eta$, the weight decay $\lambda$ differently for each layer, separately for the weights and biases, and for each iteration.

**Lemma 10.** *Fix some $\tau > 0, \delta > 0$. Let $\mathcal{S}$ be a set of $m$ examples chosen i.i.d. from $\mathcal{D}$. Then, if $m \geq \frac{4 \log(4nr/\delta)}{\tau^2}$, with probability at least $1 - \delta$ over the choice of $\mathcal{S}$, it holds that:*

$$\|\nabla_{\mathbf{W}, \mathbf{b}} L_{\mathcal{D}}(f_{\mathbf{W}, \mathbf{b}, \mathbf{u}, \beta}) - \nabla_{\mathbf{W}, \mathbf{b}} L_{\mathcal{S}}(f_{\mathbf{W}, \mathbf{b}, \mathbf{u}, \beta})\|_\infty \leq \tau$$

*Proof.* Denote by $\theta \in \mathbb{R}^{nr+r}$ the set of all parameters in $\mathbf{W}, \mathbf{b}$. For every $i \in [nr + r]$, from Hoeffding's inequality, we have:

$$\Pr \left( \left| \frac{\partial}{\partial \theta_i} L_{\mathcal{D}}(f_\theta) - \frac{\partial}{\partial \theta_i} L_{\mathcal{S}}(f_\theta) \right| \geq \tau \right) \leq 2 \exp(-m\tau^2/4) \leq \frac{\delta}{2nr}$$

and the required follows from the union bound. $\qquad \square$

Given some initialization of $\mathbf{W}, \mathbf{b}$, for every $j$ denote by $I_j \subseteq [r/2]$ the set of indices of neurons with *good* weights and bias equal to $\beta_j$. We say that an initialization is $r'$-*good* if for all $j$ we have $r'/2 \le |I_j| \le 2r'$.

Let $g$ be some vector-valued function. We define:

$$\|g\|_{\infty,2} = \sup_x \|g(x)\|_2$$

For some mapping $\psi : \mathcal{X} \to \mathbb{R}^r$ and some $B > 0$, denote by $\mathcal{H}_{\psi,B}$ the class of linear functions of norm at most $B$ over the mapping $\psi$:

$$\mathcal{H}_{\psi,B} = \{h_{\psi,\mathbf{u}} \ : \ \|\mathbf{u}\|_2 \le B\}$$

where $h_{\psi,\mathbf{u}}(\mathbf{x}) = \langle \psi(\mathbf{x}), \mathbf{u} \rangle$.

Denote by $\phi^{(t)}$ the output of the first layer of $f_{\mathbf{W},\mathbf{b},\mathbf{u},\beta}$ after $t$ iterations of GD[8].

**Lemma 11.** *Fix some $\delta > 0$. Fix some $r'$-good initialization $\mathbf{W}, \mathbf{b}$. Let $\eta = \frac{1}{2k} |C_{k,s}|^{-1}$, and let $\tau \le \frac{1}{\eta 16 kn}$. Assume that $m \ge \frac{4 \log(4nr/\delta)}{\tau^2}$. Then, there exists some mapping $\psi$ s.t. the following holds:*

1. *$\|\psi\|_{\infty,2} \le \sqrt{8kr'}$*

2. *There exists $\mathbf{u}^\star$ s.t. $L_{\mathcal{D}}(h_{\psi,\mathbf{u}^\star}) \le \frac{\sqrt{4kr'} B_k}{\sqrt{s}}$, with $\|\mathbf{u}^\star\|_2 \le B_k$ for some constant $B_k$.*

3. *With probability at least $1 - \delta$ over the choice of $\mathcal{S} \sim \mathcal{D}^m$, we have*

$$\left\| \psi - \phi^{(1)} \right\|_{\infty,2} \le 4kr'n\eta\tau$$

*Proof.* We construct two mappings $\psi, \psi^\star$ as follows.

- We will denote $\psi_0(\mathbf{x}) = \psi_0^\star(\mathbf{x}) = 1$ to allow a bias term.

- For every $i$, if $\mathbf{w}_i$ is a bad neuron, we set $\psi_i(\mathbf{x}) = \psi_i^\star(\mathbf{x}) = 0$.

- For every good neuron $\mathbf{w}_i$ s.t. $i \in I_j$, we set:

  - $\psi_i^\star(\mathbf{x}) = \sigma\left(\eta C_{k,s} \sum_{j' \in S} x_{j'} + \beta_j\right)$
  - $\psi_i(\mathbf{x}) = \sigma\left(\eta C_{k,s} \sum_{j' \in S} x_{j'} + \eta c_{k,s} \sum_{j' \in \mathbf{w}_i \setminus S} x_{j'} + \beta_j\right)$
  - $\psi_{i+r/2}^\star(\mathbf{x}) = \sigma\left(-\eta C_{k,s} x_{j'} + \beta_j\right)$
  - $\psi_{i+r/2}(\mathbf{x}) = \sigma\left(-\eta C_{k,s} \sum_{j' \in S} x_{j'} + \eta c_{k,s} \sum_{j' \in \mathbf{w}_i \setminus S} x_{j'} + \beta_j\right)$

We will show that $\psi^\star$ achieves loss zero, and that $\psi$ approximates it. We assume $C_{k,s} > 0$ and the case of $C_{k,s} < 0$ is derived similarly.

First, notice that from Lemma 8,

$$|\eta c_{k,s}| = \frac{1}{2k} \frac{|c_{k,s}|}{|C_{k,s}|} \le \frac{2}{s}$$

**Claim:** for every $\mathbf{u}$, $|L_{\mathcal{D}}(h_{\psi,\mathbf{u}}) - L_{\mathcal{D}}(h_{\psi^\star,\mathbf{u}})| \le \frac{\|\mathbf{u}\| \sqrt{4kr'}}{\sqrt{s}}$

---

[8]We assume 1 is appended to the vector for allowing bias

**Proof:** Observe that

$$|L_\mathcal{D}(h_{\psi,\mathbf{u}}) - L_\mathcal{D}(h_{\psi^\star,\mathbf{u}})| \leq \mathop{\mathbb{E}}_\mathcal{D}\left[|\ell(h_{\psi,\mathbf{u}}(\mathbf{x}),y) - \ell(h_{\psi^\star,\mathbf{u}}(\mathbf{x}),y)|\right] \leq \mathop{\mathbb{E}}_\mathcal{D}\left[|h_{\psi^\star,\mathbf{u}}(\mathbf{x}) - h_{\psi,\mathbf{u}}(\mathbf{x})|\right]$$

$$= \mathop{\mathbb{E}}_\mathcal{D}\left[|\langle \psi(\mathbf{x}) - \psi^\star(\mathbf{x}), \mathbf{u}\rangle|\right] \overset{\text{C.S}}{\leq} \|\mathbf{u}\| \mathop{\mathbb{E}}_\mathcal{D}\left[\|\psi(\mathbf{x})^\star - \psi(\mathbf{x})\|\right]$$

$$\overset{\text{Jensen}}{\leq} \|\mathbf{u}\| \sqrt{\mathop{\mathbb{E}}_\mathcal{D}\left[\|\psi(\mathbf{x})^\star - \psi(\mathbf{x})\|^2\right]} = \|\mathbf{u}\| \sqrt{\sum_i \mathop{\mathbb{E}}_\mathcal{D}\left[(\psi_i^\star(\mathbf{x}) - \psi_i(\mathbf{x}))^2\right]}$$

$$\leq \|\mathbf{u}\| \sqrt{\sum_{i \text{ is good}} \mathop{\mathbb{E}}_\mathcal{D}\left[\left(\eta c_{k,s} \sum_{j' \in \mathbf{w}_i \setminus S} x_{j'}\right)^2\right]} \leq \|\mathbf{u}\| \sqrt{4kr'} |\eta c_{k,s}| \sqrt{s}$$

$$\leq \frac{\|\mathbf{u}\| \sqrt{4kr'}}{\sqrt{s}}$$

**Claim:** There exists $\mathbf{u}^\star$ with norm $\|\mathbf{u}^\star\| \leq B_k$ and $L_\mathcal{D}(h_{\psi^\star,\mathbf{u}^\star}) = 0$.

**Proof:** For every $\mathbf{x}$, denote $s_\mathbf{x} = \sum_{j \in S} x_j$, and observe that $s_\mathbf{x} \in \{-k, -k+2, \ldots, k-2, k\} =: \mathcal{S}$ and $\chi_S(\mathbf{x}) = s_\mathbf{x} \mod 2$. For every $j$, denote $v_j^+(s) = \sigma(\frac{1}{2k}s + \beta_j)$ and $v_j^-(s) = \sigma(-\frac{1}{2k}s + b_j)$. Then, for every $s \in \mathcal{S}$ denote $\mathbf{v}(s) = (1, v_1^+(s), \ldots, v_{k/2-1}^+(s), v_1^-(s), \ldots, v_{k/2-1}^-(s)) \in \mathbb{R}^k$. Observe that $V = \{\mathbf{v}(s)\}_{s \in \mathcal{S}} \in \mathbb{R}^{k \times k}$ has linearly independent rows, and therefore there exists $\nu = (\nu_0, \nu_1^+, \ldots, \nu_{k/2-1}^+, \nu_1^-, \ldots, \nu_{k/2-1}^-) \in \mathbb{R}^k$ s.t. $\mathbf{v}(s)^\top \nu = s \mod 2$. Now, define $\mathbf{u}^\star$ s.t. for every bad $i$ we set $u_i^\star = 0$, and for every $i \in I_j$ we set $u_i^\star = \frac{1}{|I_j|}\nu_j^+$ and $u_{i+r/2,-}^\star = \frac{1}{|I_j|}\nu_j^-$, and set $u_0^\star = \nu_0$. Observe that for every $\mathbf{x} \in \mathcal{X}$ we have:

$$\langle \psi(\mathbf{x}), \mathbf{u}^\star \rangle = \nu_0 + \sum_j \frac{1}{|I_j|} \sum_{i \in I_j} (v_j^+(s_\mathbf{x})\nu_j^+ + v_j^-(s_\mathbf{x})\nu_j^-) = \langle \mathbf{v}(s_\mathbf{x})^\top \nu \rangle = s_\mathbf{x} \mod 2 = \chi_S(\mathbf{x})$$

and therefore $L_\mathcal{D}(h_{\psi,\mathbf{u}^\star}) = 0$. Additionally, observe that

$$\|\mathbf{u}^\star\|^2 = \nu_0^2 + \sum_j \frac{1}{|I_j|^2}((\nu_j^+)^2 + (\nu_j^-)^2) \leq \|\nu\|_2^2$$

Now we prove the statements in the main lemma:

1. For every $\mathbf{x} \in \mathcal{X}$ we have

$$\|\psi(\mathbf{x})\|_2^2 = \sum_j \sum_{i \in I_j} (\psi_i(\mathbf{x})^2 + \psi_{i+r/2}(\mathbf{x})^2) \leq 4 \sum_j |I_j| \leq 8kr'$$

2. Follows from the two previous claims.

3. Assume we choose $\lambda = 1$ for the weights of the first layer, $\lambda = 0$ for the biases of the first layer, $\eta = \frac{1}{2k}|C_{k,s}|^{-1}$ for the weights of the first layer, and $\eta = 0$ for all other parameters. From Lemma 10, w.p. at least $1 - \delta$ we have:

$$\|\nabla_{\mathbf{W},\mathbf{b}}L_\mathcal{D}(f_{\mathbf{W},\mathbf{b},\mathbf{u},\beta}) - \nabla_{\mathbf{W},\mathbf{b}}L_\mathcal{S}(f_{\mathbf{W},\mathbf{b},\mathbf{u},\beta})\|_\infty \leq \tau$$

Denote by $\mathbf{w}_i^{(1)}$ the $i$-th weight after the first gradient step, and denote $\mathbf{w}_i^\star := -\eta \nabla_{\mathbf{w}_i}L_\mathcal{D}(f_{\mathbf{W},\mathbf{b},\mathbf{u},\beta})$ and $\widehat{\mathbf{w}}_i := -\eta \nabla_{\mathbf{w}_i}L_\mathcal{S}(f_{\mathbf{W},\mathbf{b},\mathbf{u},\beta})$. By the choice of $\lambda$, we get $\mathbf{w}_i^{(1)} = \widehat{\mathbf{w}}_i$. Observe that for all $i$:

$$\|\widehat{\mathbf{w}}_i - \mathbf{w}_i^\star\|_\infty = \eta \|\nabla_{\mathbf{w}_i}L_\mathcal{S}(f_{\mathbf{W},\mathbf{b},\mathbf{u},\beta}) - \nabla_{\mathbf{w}_i}L_\mathcal{D}(f_{\mathbf{W},\mathbf{b},\mathbf{u},\beta})\| \leq \eta\tau$$

**Claim:** For all $i, j$, if $w_{i,j}^\star = 0$, then $\left|w_{i,j}^\star - w_{i,j}^{(1)}\right| \leq \frac{1}{16kn}$.

**Proof:** We have $|\widehat{w}_{i,j}| \leq \eta\tau$, and the claim follows from the fact that $\eta\tau \leq \frac{1}{16kn}$.

First, consider the case where $\mathbf{w}_i$ is a *bad* neuron. In this case, by Lemma 9, we have $\|\mathbf{w}_i^\star\|_1 \le \frac{1}{2k}$ and $\|\mathbf{w}_i^\star\|_0 \le 1$, and from the previous claim we get $\left\|\mathbf{w}_i^{(1)} - \mathbf{w}_i^\star\right\|_1 \le \eta\tau + \frac{1}{16k}$. Therefore, for all $\mathbf{x} \in \mathcal{X}$ we get:

$$\left|\left\langle \mathbf{w}_i^{(1)}, \mathbf{x}\right\rangle\right| = \left|\left\langle \mathbf{w}_i^{(1)} - \mathbf{w}_i^\star + \mathbf{w}_i^\star, \mathbf{x}\right\rangle\right| \le \left(\|\mathbf{w}_i^\star\|_1 + \left\|\mathbf{w}_i^\star - \mathbf{w}_i^{(1)}\right\|_1\right)\|\mathbf{x}\|_\infty \le \frac{10}{16k} + \eta\tau$$

Since at initialization we have $b_i \le -\frac{15}{16k}$, and we have $\eta\tau \le \frac{1}{16k}$, for all $\mathbf{x} \in \mathcal{X}$:

$$\sigma\left(\left\langle \mathbf{w}_i^{(1)}, \mathbf{x}\right\rangle + b_i\right) = 0 = \psi_i(\mathbf{x})$$

Now, assume that $\mathbf{w}_i$ is a *good* neuron. In this case, by Lemma 8, we have

$$\mathbf{w}_{i,j}^\star = \begin{cases} \eta C_{k,s} & j \in S \\ w_{i,j}\eta c_{k,s} & j \notin S \end{cases}$$

Observe that $\psi_i(\mathbf{x}) = \sigma\left(\langle \mathbf{w}_i^\star, \mathbf{x}\rangle + b_i\right)$, and therefore:

$$\left|\psi_i(\mathbf{x}) - \sigma\left(\left\langle \mathbf{w}_i^{(1)}, \mathbf{x}\right\rangle + b_i\right)\right| \le \left|\left\langle \mathbf{w}_i^{(1)} - \mathbf{w}_i^\star, \mathbf{x}\right\rangle\right| \le \left\|\mathbf{w}_i^{(1)} - \mathbf{w}_i^\star\right\|_1 \le n\eta\tau$$

Similarly, in this case we will get $\left|\psi_{i+r/2}(\mathbf{x}) - \sigma\left(\left\langle \mathbf{w}_{i+r/2}^{(1)}, \mathbf{x}\right\rangle + b_{i+r/2}\right)\right| \le n\eta\tau$. Now the required follows from all we showed.

$\square$

**Lemma 12.** *Fix some mappings $\psi, \psi'$ and some $\mathbf{w}$. Then, for for every distribution $\mathcal{D}$:*

$$|L_\mathcal{D}(h_{\psi,\mathbf{w}}) - L_\mathcal{D}(h_{\psi',\mathbf{w}})| \le \|\mathbf{w}\|\,\|\psi - \psi'\|_{\infty,2}$$

*and for every sample $\mathcal{S}$:*

$$|L_\mathcal{S}(h_{\psi,\mathbf{w}}) - L_\mathcal{S}(h_{\psi',\mathbf{w}})| \le \|\mathbf{w}\|\,\|\psi - \psi'\|_{\infty,2}$$

*Proof.* Observe that, since $\ell$ is 1-Lipschitz:

$$|L_\mathcal{D}(h_{\psi',\mathbf{w}}) - L_\mathcal{D}(h_{\psi,\mathbf{w}})| \le \mathop{\mathbb{E}}_{(\mathbf{x},y)\sim\mathcal{D}}\left[|\ell(h_{\psi',\mathbf{w}}(\mathbf{x}), y) - \ell(h_{\psi',\mathbf{w}}(\mathbf{x}), y)|\right]$$

$$\le \mathop{\mathbb{E}}_\mathcal{D}\left[|h_{\psi',\mathbf{w}}(\mathbf{x}) - h_{\psi,\mathbf{w}}(\mathbf{x})|\right] \le \mathop{\mathbb{E}}_\mathcal{D}\left[|\langle\psi'(\mathbf{x}) - \psi(\mathbf{x}), \mathbf{w}\rangle|\right]$$

$$\le \mathop{\mathbb{E}}_\mathcal{D}\left[\|\psi'(\mathbf{x}) - \psi(\mathbf{x})\|\,\|\mathbf{w}\|\right] \le \|\mathbf{w}\|\,\|\psi - \psi'\|_{\infty,2}$$

and similarly we get:

$$|L_\mathcal{S}(h_{\psi',\mathbf{w}}) - L_\mathcal{S}(h_{\psi,\mathbf{w}})| \le \|\mathbf{w}\|\,\|\psi - \psi'\|_{\infty,2}$$

$\square$

**Lemma 13.** *Fix some mapping $\psi$, and let $\mathcal{S}$ be a sample of size $m$ sampled i.i.d. from $\mathcal{D}$. Then, with probability at least $1 - \delta$ over the choice of $\mathcal{S}$, for every $\psi'$ and for every $h \in \mathcal{H}_{\psi',B}$, we have:*

$$L_\mathcal{D}(h) \le L_\mathcal{S}(h) + \frac{(2B\,\|\psi\|_{\infty,2} + 1)\sqrt{2\log(2/\delta)}}{\sqrt{m}} + 2B\,\|\psi - \psi'\|_{\infty,2}$$

*Proof.* First, observe that using Theorem 26.12 in Shalev-Shwartz and Ben-David (2014), with probability at least $1 - \delta$ over the choice of $\mathcal{S}$, for every $h_{\psi,\mathbf{w}} \in \mathcal{H}_{\psi,B}$ we have:

$$L_\mathcal{D}(h_{\psi,\mathbf{w}}) \le L_\mathcal{S}(h_{\psi,\mathbf{w}}) + \frac{(2B\,\|\psi\|_{\infty,2} + 1)\sqrt{2\log(2/\delta)}}{\sqrt{m}}$$

In this case, using the previous lemma, for every $\psi'$ and every $h_{\psi',\mathbf{w}} \in \mathcal{H}_{\psi',B}$ we have:

$$L_{\mathcal{D}}(h_{\psi',\mathbf{w}}) \leq L_{\mathcal{D}}(h_{\psi,\mathbf{w}}) + B\left\|\psi - \psi'\right\|_{\infty,2}$$

$$\leq L_{\mathcal{S}}(h_{\psi,\mathbf{w}}) + \frac{(2B\|\psi\|_{\infty,2} + 1)\sqrt{2\log(2/\delta)}}{\sqrt{m}} + B\left\|\psi - \psi'\right\|_{\infty,2}$$

$$\leq L_{\mathcal{S}}(h_{\psi',\mathbf{w}}) + \frac{(2B\|\psi\|_{\infty,2} + 1)\sqrt{2\log(2/\delta)}}{\sqrt{m}} + 2B\left\|\psi - \psi'\right\|_{\infty,2}$$

$\square$

**Lemma 14.** *Fix $\epsilon, \delta \in (0, 1/2)$. Let $\psi$ be some mapping s.t. there exists $\mathbf{w}^\star$ satisfying $\|\mathbf{w}^\star\| \leq B$ and $L_{\mathcal{D}}(h_{\psi,\mathbf{w}^\star}) \leq \epsilon$. Let $\mathcal{S}$ be a sample of size $m$ from $\mathcal{D}$. With probability at least $1 - 2\delta$ over the choice of $\mathcal{S}$, there exists a choice of learning rate, weight decay and truncation parameters s.t. if $\left\|\phi^{(1)} - \psi\right\|_{\infty,2} \leq \frac{\epsilon}{B}$ and $\frac{T}{\log(T)} \geq \frac{100}{\epsilon^2}\left(\|\psi\|_{\infty,2} + B^{-1}\right)^2$ and $m \geq \frac{(4\sqrt{2}B\|\psi\|_{\infty,2}+1)^2 \log(2/\delta)}{\epsilon^2}$, GD returns a function $h$ s.t. $L_{\mathcal{D}}(h) \leq 7\epsilon$.*

*Proof.* Consider the following convex function:

$$L(\mathbf{w}) = L_{\mathcal{S}}(h_{\phi^{(1)},\mathbf{w}}) + \frac{\lambda}{2}\|\mathbf{w}\|^2$$

**Claim 1**: for all $\mathbf{x}$ and $y$, we have $|\ell(h_{\psi,\mathbf{w}^\star}(\mathbf{x}), y)| \leq B\|\psi\|_{\infty,2}$.

**Proof**: Observe that,

$$|\ell(h_{\psi,\mathbf{w}^\star}(\mathbf{x}), y)| \leq |h_{\psi,\mathbf{w}^\star}(\mathbf{x})| = |\langle\psi(\mathbf{x}), \mathbf{w}^\star\rangle| \leq \|\psi(\mathbf{x})\|_2 \|\mathbf{w}^\star\| \leq B\|\psi\|_{\infty,2}$$

**Claim 2**: W.p. at least $1 - \delta$ we have $L_{\mathcal{S}}(h_{\psi,\mathbf{w}^\star}) \leq \epsilon + \frac{B\|\psi\|_{\infty,2}\sqrt{2\log(2/\delta)}}{\sqrt{m}}$

**Proof**: from Hoeffding's inequality, using the previous claim:

$$\Pr\left[L_S(h) \geq L_{\mathcal{D}}(h) + t\right] \leq \exp\left(-\frac{mt^2}{2\|\psi\|_{\infty,2}^2 B^2}\right)$$

And therefore,

$$\Pr\left[L_{\mathcal{S}}(h_{\psi,\mathbf{w}^\star}) \geq L_{\mathcal{D}}(h_{\psi,\mathbf{w}^\star}) + \frac{\|\psi\|_{\infty,2}B\sqrt{2\log(2/\delta)}}{\sqrt{m}}\right] \leq \delta/2$$

and the required follows from the assumption $L_{\mathcal{D}}(h_{\psi,\mathbf{w}^\star}) \leq \epsilon$

**Claim 3**: $L(\mathbf{w}^\star) \leq 2\epsilon + \frac{B\|\psi\|_{\infty,2}\sqrt{2\log(2/\delta)}}{\sqrt{m}} + \frac{\lambda B^2}{2}$.

**Proof**: from the previous claim, we have $L_{\mathcal{S}}(h_{\psi,\mathbf{w}^\star}) \leq \epsilon + \frac{B\|\psi\|_{\infty,2}\sqrt{2\log(2/\delta)}}{\sqrt{m}}$. Using Lemma 12, we get $L_{\mathcal{S}}(h_{\phi^{(1)},\mathbf{w}^\star}) \leq \epsilon + \frac{B\|\psi\|_{\infty,2}\sqrt{2\log(2/\delta)}}{\sqrt{m}} + B\left\|\psi - \phi^{(1)}\right\|_{\infty,2}$ and the required follows.

**Claim 3**: there exists a step-size schedule for GD s.t. $L_{\mathcal{S}}(\mathbf{w}_T) \leq \inf_{\mathbf{w}} L_{\mathcal{S}}(\mathbf{w}) + \frac{100(\|\psi\|_{\infty,2}+\epsilon/B)^2(1+\log(T))}{\lambda T}$.

**Proof**: Using Shamir and Zhang (2013)

Combining the previous claims, we get:

$$L(\mathbf{w}_T) \leq 2\epsilon + \frac{B\|\psi\|_{\infty,2}\sqrt{2\log(2/\delta)}}{\sqrt{m}} + \frac{\lambda B^2}{2} + \frac{100(\|\psi\|_{\infty,2} + \epsilon/B)^2(1 + \log(T))}{\lambda T}$$

Now, choosing $\lambda = \frac{\epsilon}{B^2}$ we get:

$$L(\mathbf{w}_T) \leq 2\epsilon + \frac{B\|\psi\|_{\infty,2}\sqrt{2\log(2/\delta)}}{\sqrt{m}} + \frac{100B^2(\|\psi\|_{\infty,2} + \epsilon/B)^2(1 + \log(T))}{\epsilon T}$$

So, if $\frac{T}{\log(T)} \geq \frac{100}{\epsilon^2}\left(\|\psi\|_{\infty,2} + B^{-1}\right)^2$ and $m \geq \frac{(4\sqrt{2}B\|\psi\|_{\infty,2}+1)^2\log(2/\delta)}{\epsilon^2}$ we have $L(\mathbf{w}_T) \leq 4\epsilon$ and therefore $\|\mathbf{w}_T\|^2 \leq \frac{2}{\lambda}L(\mathbf{w}_T) \leq 8B^2$.

In this case, using Lemma 13, we w.p. at least $1 - \delta$:

$$L_\mathcal{D}(h_{\phi^{(1)},\mathbf{w}^T}) \leq L_\mathcal{S}(h_{\phi^{(1)},\mathbf{w}^T}) + \frac{(4\sqrt{2}B\|\psi\|_{\infty,2}+1)\sqrt{2\log(2/\delta)}}{\sqrt{m}} + 2B\left\|\psi - \phi^{(1)}\right\|_{\infty,2} \leq 7\epsilon$$

$\square$

**Lemma 15.** *Fix some $\delta$. Assume we initialize a network of size $r \geq 20k(2n/s)^k \log\left(\frac{2k}{\delta}\right)$. Then, w.p. at least $1 - \delta$, $\mathbf{W}, \mathbf{b}$ is $r'$-good for $r' = \frac{r}{2k}(s/2n)^k$.*

*Proof.* From Lemma 7, the probability of drawing a good neuron is $\geq (s/2n)^k$. So, for every $i$, the probability of drawing a *good* neuron with bias $\beta_i$ is at least $\frac{1}{k}(s/2n)^k$. Denote by $r'_i$ the number of good neurons with bias $\beta_i$. Observe that $\mathbb{E}[r'_i] = \frac{r}{2k}(s/2n)^k$. Using Chernoff's bound, we have:

$$\Pr\left[r'_i > \frac{r}{4k}(s/2n)^k\right] \leq \exp\left(-\frac{r}{20k}(s/2n)^k\right) \leq \frac{\delta}{2k}$$

and similarly $\Pr\left[r'_i < \frac{3r}{4k}(s/2n)^k\right] \leq \frac{\delta}{2k}$. So, using the union bound we get the required. $\square$

**Theorem 16.** *Fix $\delta \in (0, 1/2), \epsilon \in (0, 1/2)$, and assume we choose:*

- $s \geq \alpha_1^{(k)} \frac{\log(1/\delta)}{\epsilon^2}$.

- $r = \left\lceil \alpha_2^{(k)}(n/s)^k \log(1/\delta) \right\rceil$

- $m \geq \alpha_3^{(k)}\binom{s}{k-1}n^2 \log(nr/\delta)\log(1/\delta)$

- $T \geq \alpha_4^{(k)} \frac{\log(1/\delta)\log(T)}{\epsilon^2}$

*for some constants $\alpha_1^{(k)}, \alpha_2^{(k)}, \alpha_3^{(k)}, \alpha_4^{(k)}$. Then, with probability at least $1 - \delta$ over the choice of sample size and initialization, gradient descent returns after $T$ iterations a function $h$ s.t. $L_\mathcal{D}(h) \leq \epsilon$.*

*Proof.* Choose $r = \left\lceil 20k(2n/s)^k \log\left(\frac{2k}{\delta}\right)\right\rceil$.

- From Lemma 15, with probability at least $1 - \delta$ we get an $r'$-*good* init with $r' = \frac{r}{2k}(s/2n)^k$ and notice that $10\log(2k/\delta) \leq r' \leq 20\log(2k/\delta)$.

- Let $\eta = \frac{1}{2k}|C_{k,s}|^{-1} \geq \frac{\kappa_k}{2k}\sqrt{\binom{s}{k-1}}$ and

$$\tau = \frac{\epsilon}{40B_k n\kappa_k \sqrt{\binom{s}{k-1}}\log(2k/\delta)} \leq \min\left(\frac{\epsilon}{4B_k kr'n\eta}, \frac{1}{16\eta kn}\right)$$

  Choosing

  - $m \geq \frac{4 \cdot 40^2 B_k^2 \kappa_k n^2 \binom{s}{k-1}\log(2k/\delta)^2 \log(4nr/\delta)}{\epsilon^2} \geq \frac{4\log(4nr/\delta)}{\tau^2}$
  - $s \geq \frac{80kB_k^2 \log(2k/\delta)}{\epsilon^2} \geq \frac{4kr' B_k^2}{\epsilon^2}$

  from Lemma 11, the conditions for Lemma 14 are satisfied with

  1. $B = B_k$.
  2. $\|\psi\|_{\infty,2} \leq \sqrt{8kr'} \leq 4\sqrt{10k\log(2k/\delta)}$
  3. $\left\|\psi - \phi^{(1)}\right\|_{\infty,2} \leq 4kr'n\eta\tau \leq \frac{\epsilon}{B_k}$

- Choosing

$$- \frac{T}{\log(T)} \geq \frac{100\left(4\sqrt{10k\log(2k/\delta)}+B^{-1}\right)^2}{\epsilon^2} \geq \frac{100}{\epsilon^2}\left(\|\psi\|_{\infty,2}+B^{-1}\right)^2$$

$$- m \geq \frac{(16\sqrt{2}B\sqrt{10k\log(2k/\delta)}+1)^2\log(2/\delta)}{\epsilon^2} \geq \frac{(4\sqrt{2}B\|\psi\|_{\infty,2}+1)^2\log(2/\delta)}{\epsilon^2}$$

using Lemma 14 we get w.p. at least $1-2\delta$ we have $L_{\mathcal{D}}(h) \leq 7\epsilon$.

$\square$

### B.3 Feature selection with an under-sparse initialization and a narrow network

Fix some subset $S \subseteq \binom{n}{k}$ to be the true parity function. Let $k$ be even.

We train the following network:

$$f_{\mathbf{W},\mathbf{b},\mathbf{u},\beta}(\mathbf{x}) = \sum_{i=1}^{r} u_i \sigma\left(\langle \mathbf{w}_i, \mathbf{x}\rangle + b_i\right) + \beta.$$

We initialize the network as follows:

- Randomly initialize $\mathbf{w}_1, \ldots, \mathbf{w}_{r/2} \in \{\epsilon, 1\}^n$ s.t. $s$ coordinates have weight 1 and rest have weight $\epsilon < \frac{1}{(n-s)}$, with a uniform distribution over all $\binom{n}{s}$ subsets.

- Randomly initialize $b_1, \ldots, b_{r/2} \sim \left\{-\epsilon\frac{k-1}{2k}, -\epsilon\frac{k-3}{2k}, \ldots, -\epsilon\frac{1}{2k}, \epsilon\frac{1}{2k}, \ldots, \epsilon\frac{k-3}{2k}, \epsilon\frac{k-1}{2k}\right\}$.

- Randomly initialize $u_1, \ldots, u_{r/2} \in \{\pm 1\}$ uniformly at random.

- Initialize $\mathbf{w}_{r/2+1}, \ldots, \mathbf{w}_r, b_{r/2+1}, \ldots, b_r$ s.t. $\mathbf{w}_i = \mathbf{w}_{i-r/2}$ and $b_i = b_{i-r/2}$ (symmetric initialization).

- Initialize $u_{r/2+1}, \ldots, u_r$ s.t. $u_i = -u_{i-r/2}$.

- Initialize $\beta = 0$

Similar to the over-sparse case, we consider hinge-loss. We consider one-step of gradient descent on a sample $\mathcal{S}$ with $\ell_2$ regularization (weight decay):

$$\theta^{(1)} = (1-\lambda)\theta^{(0)} - \eta \operatorname{trunc}(\nabla L_{\mathcal{S}}(f_{\theta^{(0)}}), \gamma)$$

with learning rate $\eta$, truncation parameter $\gamma$, and the weight decay $\lambda$ chosen differently for each layer and separately for the weights and biases. Note that truncation just zeros out gradients with magnitude $\leq \gamma$.

**Majority and Half.** We will make use of two Boolean functions: (1) Majority, and (2) Half (derivative of Majority). For input $x \in \{\pm 1\}^n$, we define

$$\operatorname{Maj}(x) = \operatorname{sign}\left(\sum_{i=1}^{n} x_i\right)$$

where $\operatorname{sign}(a) = 1$ is $a > 0$ else $-1$. The derivative of the Majority function is denoted by $D_n\operatorname{Maj}_n = \operatorname{Half}$ and is defined for input $x \in \{\pm 1\}^{n-1}$ as:

$$\operatorname{Half}(x) = \mathbb{1}\left(\sum_{i=1}^{n} x_i = 0\right).$$

The corresponding Fourier coefficients corresponding to set $S$ are denoted by $\widehat{\operatorname{Maj}}_n(S)$ and $\widehat{\operatorname{Half}}_{n-1}(S)$. Note that both functions are permutation invariant, so the Fourier coefficients only depend on the size of the set.

**Lemma 17** (O'Donnell (2014)). *For any integers $m \geq j$, we have*

$$\widehat{\operatorname{Half}}_{2m}(2j) = \widehat{\operatorname{Maj}}_{2m+1}(2j+1) = (-1)^j \frac{\binom{m}{j}}{\binom{2m}{2j}}\frac{\binom{2m}{m}}{2^{2m}}.$$

**Population gradients at initialization.** Consider a fixed weight $\mathbf{w}$ with the set of 1 weights indicated by $S' \subseteq S$. We first start with computing the population gradients for this neuron $(h_{\mathbf{w}}(\mathbf{x}) = \sigma(\langle \mathbf{w}, \mathbf{x} \rangle)$ be a single ReLU neuron where $\sigma(x) = \max\{x, 0\})$ at initialization for varying $S \cap S' = \bar{S}$ and parity and non-parity variables.

**Lemma 18** (Population gradient at initialization for parity variables). *Assuming $s < k$ with $s, k$ even, for $i \in S$, we have*

$$\underset{\mathbf{x} \sim \{\pm 1\}^n}{\mathbb{E}} \left[ \frac{\partial}{\partial w_i} h_{\mathbf{w}}(\mathbf{x}) \cdot \chi_S(\mathbf{x}) \right] = \frac{1}{2} \widehat{\mathrm{Half}}_s(\bar{S} \setminus \{i\}) \widehat{\mathrm{Maj}}_{n-s}(S \setminus (\bar{S} \cup \{i\})).$$

*Proof.* For $i \in S$, we have

$$\underset{\mathbf{x} \sim \{\pm 1\}^n}{\mathbb{E}} \left[ \frac{\partial}{\partial w_i} h_{\mathbf{w}}(\mathbf{x}) \cdot \chi_S(\mathbf{x}) \right]$$

$$= \underset{\mathbf{x} \sim \{\pm 1\}^n}{\mathbb{E}} \left[ \sigma'(\langle \mathbf{w}, \mathbf{x} \rangle) \cdot x_i \cdot \chi_S(\mathbf{x}) \right]$$

Since $i \in S$:

$$= \underset{\mathbf{x} \sim \{\pm 1\}^n}{\mathbb{E}} \left[ \mathbf{1}_{\{\sum_{j \in S'} x_j + \epsilon \sum_{j \notin S'} x_j > 0\}} \chi_{S \setminus \{i\}}(\mathbf{x}) \right]$$

Splitting based on value of $\sum_{j \in S'} x_j$:

$$= \underset{\mathbf{x} \sim \{\pm 1\}^n}{\mathbb{E}} \left[ \mathbf{1}_{\{\sum_{j \in S'} x_j = 0\}} \mathbf{1}_{\{\sum_{j \notin S'} x_j > 0\}} \chi_{S \setminus \{i\}}(\mathbf{x}) \right] + \underset{\mathbf{x} \sim \{\pm 1\}^n}{\mathbb{E}} \left[ \mathbf{1}_{\{\sum_{j \in S'} x_j > 0\}} \mathbf{1}_{\{\sum_{j \in S'} x_j + \epsilon \sum_{j \notin S'} x_j > 0\}} \chi_{S \setminus \{i\}}(\mathbf{x}) \right]$$

Using the fact that $\left| \epsilon \sum_{j \notin S'} x_j \right| \leq (n - s)\epsilon < 1$:

$$= \underset{\mathbf{x} \sim \{\pm 1\}^n}{\mathbb{E}} \left[ \mathbf{1}_{\{\sum_{j \in S'} x_j = 0\}} \mathbf{1}_{\{\sum_{j \notin S'} x_j \geq 0\}} \chi_{S \setminus \{i\}}(\mathbf{x}) \right] + \underset{\mathbf{x} \sim \{\pm 1\}^n}{\mathbb{E}} \left[ \mathbf{1}_{\{\sum_{j \in S'} x_j > 0\}} \chi_{S \setminus \{i\}}(\mathbf{x}) \right]$$

Splitting between variables in $S'$ and outside $S'$:

$$= \underset{\mathbf{x} \sim \{\pm 1\}^n}{\mathbb{E}} \left[ \mathbf{1}_{\{\sum_{j \in S'} x_j = 0\}} \chi_{\bar{S} \setminus \{i\}}(\mathbf{x}) \right] \underset{\mathbf{x} \sim \{\pm 1\}^n}{\mathbb{E}} \left[ \mathbf{1}_{\{\sum_{j \notin S'} x_j \geq 0\}} \chi_{S \setminus (\bar{S} \cup \{i\})}(\mathbf{x}) \right] + \underset{\mathbf{x} \sim \{\pm 1\}^n}{\mathbb{E}} \left[ \mathbf{1}_{\{\sum_{j \in S'} x_j > 0\}} \chi_{S \setminus \{i\}}(\mathbf{x}) \right]$$

Replacing indicators with Maj and Half appropriately:

$$= \underset{\mathbf{x} \sim \{\pm 1\}^n}{\mathbb{E}} \left[ \mathrm{Half}_s(\mathbf{x}_{S'}) \chi_{\bar{S} \setminus \{i\}}(\mathbf{x}) \right] \underset{\mathbf{x} \sim \{\pm 1\}^n}{\mathbb{E}} \left[ \frac{1}{2} \left( \mathrm{Maj}_{n-s}(\mathbf{x}_{[n] \setminus S'}) + 1 \right) \chi_{S \setminus (\bar{S} \cup \{i\})}(\mathbf{x}) \right]$$

$$+ \underset{\mathbf{x} \sim \{\pm 1\}^n}{\mathbb{E}} \left[ \frac{1}{2} \left( \mathrm{Maj}_s(\mathbf{x}_{S'}) + 1 \right) \chi_{\bar{S} \setminus \{i\}}(\mathbf{x}) \right] \underset{\mathbf{x} \sim \{\pm 1\}^{n-s}}{\mathbb{E}} \left[ \chi_{S \setminus (\bar{S} \cup \{i\})}(\mathbf{x}) \right]$$

Replacing indicators with Fourier coefficients appropriately:

$$= \frac{1}{2} \widehat{\mathrm{Half}}_s(\bar{S} \setminus \{i\}) \left( \widehat{\mathrm{Maj}}_{n-s}(S \setminus (\bar{S} \cup \{i\})) + \mathbf{1}_{\{S \subseteq \bar{S} \cup \{i\}\}} \right) + \frac{1}{2} \left( \widehat{\mathrm{Maj}}_s(\bar{S} \setminus \{i\}) + \mathbf{1}_{\{\bar{S} \setminus \{i\} = \phi\}} \right) \mathbf{1}_{\{S \subseteq \bar{S} \cup \{i\}\}}$$

$$= \frac{1}{2} \widehat{\mathrm{Half}}_s(\bar{S} \setminus \{i\}) \widehat{\mathrm{Maj}}_{n-s}(S \setminus (\bar{S} \cup \{i\})) + \frac{1}{2} \left( \widehat{\mathrm{Maj}}_s(\bar{S} \setminus \{i\}) + \mathbf{1}_{\{\bar{S} \setminus \{i\} = \phi\}} + \widehat{\mathrm{Half}}_s(\bar{S} \setminus \{i\}) \right) \mathbf{1}_{\{S \subseteq \bar{S} \cup \{i\}\}}$$

Since $k, s$ are even and $k < s$, $|\bar{S} \cup \{i\}| \leq s + 1 < k = |S|$ therefore $\mathbf{1}_{\{S \subseteq \bar{S} \cup \{i\}\}} = 0$:

$$= \frac{1}{2} \widehat{\mathrm{Half}}_s(\bar{S} \setminus \{i\}) \widehat{\mathrm{Maj}}_{n-s}(S \setminus (\bar{S} \cup \{i\})).$$

This gives us the desired result. $\qquad\square$

**Lemma 19** (Population gradient at initialization for non-parity variables). *Assuming $s < k$ with $s, k$ even, for $i \notin S$, we have*

$$\underset{\mathbf{x} \sim \{\pm 1\}^n}{\mathbb{E}} \left[ \frac{\partial}{\partial w_i} h_{\mathbf{w}}(\mathbf{x}) \cdot \chi_S(\mathbf{x}) \right] = \frac{1}{2} \widehat{\mathrm{Half}}_s(\bar{S} \cup (S' \cap \{i\})) \widehat{\mathrm{Maj}}_{n-s}((S \cup \{i\}) \setminus S').$$

*Proof.* For $i \notin S$, using similar calculations as in the proof of Lemma 18, we have

$$\mathop{\mathbb{E}}_{\mathbf{x}\sim\{\pm 1\}^n}\left[\frac{\partial}{\partial w_i}h_{\mathbf{w}}(\mathbf{x})\cdot\chi_S(\mathbf{x})\right]$$

$$=\mathop{\mathbb{E}}_{\mathbf{x}\sim\{\pm 1\}^n}\left[\sigma'(\langle\mathbf{w},\mathbf{x}\rangle)\cdot x_i\cdot\chi_S(\mathbf{x})\right]$$

$$=\mathop{\mathbb{E}}_{\mathbf{x}\sim\{\pm 1\}^n}\left[\mathbf{1}_{\{\sum_{j\in S'} x_j+\epsilon\sum_{j\notin S'} x_j>0\}}\chi_{S\cup\{i\}}(\mathbf{x})\right]$$

$$=\mathop{\mathbb{E}}_{\mathbf{x}\sim\{\pm 1\}^n}\left[\mathbf{1}_{\{\sum_{j\in S'} x_j=0\}}\mathbf{1}_{\{\sum_{j\notin S'} x_j>0\}}\chi_{S\cup\{i\}}(\mathbf{x})\right] + \mathop{\mathbb{E}}_{\mathbf{x}\sim\{\pm 1\}^n}\left[\mathbf{1}_{\{\sum_{j\in S'} x_j>0\}}\chi_{S\cup\{i\}}(\mathbf{x})\right]$$

$$=\mathop{\mathbb{E}}_{\mathbf{x}\sim\{\pm 1\}^n}\left[\mathbf{1}_{\{\sum_{j\in S'} x_j=0\}}\chi_{\bar{S}\cup(S'\cap\{i\})}(\mathbf{x})\right]\mathop{\mathbb{E}}_{\mathbf{x}\sim\{\pm 1\}^n}\left[\mathbf{1}_{\{\sum_{j\notin S'} x_j>0\}}\chi_{(S\cup\{i\})\setminus S'}(\mathbf{x})\right]$$

$$+\mathop{\mathbb{E}}_{\mathbf{x}\sim\{\pm 1\}^n}\left[\mathbf{1}_{\{\sum_{j\in S'} x_j>0\}}\chi_{S\cup\{i\}}(\mathbf{x})\right]$$

$$=\mathop{\mathbb{E}}_{\mathbf{x}\sim\{\pm 1\}^n}\left[\mathrm{Half}_s(\mathbf{x}_{S'})\chi_{\bar{S}\cup(S'\cap\{i\})}(\mathbf{x})\right]\mathop{\mathbb{E}}_{\mathbf{x}\sim\{\pm 1\}^n}\left[\frac{1}{2}\left(\mathrm{Maj}_{n-s}(\mathbf{x}_{[n]\setminus S'})+1\right)\chi_{(S\cup\{i\})\setminus S'}(\mathbf{x})\right]$$

$$+\mathop{\mathbb{E}}_{\mathbf{x}\sim\{\pm 1\}^n}\left[\frac{1}{2}\left(\mathrm{Maj}_s(\mathbf{x}_{S'})+1\right)\chi_{\bar{S}\cup(S'\cap\{i\})}(\mathbf{x})\right]\mathop{\mathbb{E}}_{\mathbf{x}\sim\{\pm 1\}^n}\left[\chi_{(S\cup\{i\})\setminus S'}(\mathbf{x})\right]$$

$$=\frac{1}{2}\widehat{\mathrm{Half}}_s(\bar{S}\cup(S'\cap\{i\}))\widehat{\mathrm{Maj}}_{n-s}((S\cup\{i\})\setminus S')$$

$$+\frac{1}{2}\left(\widehat{\mathrm{Maj}}_s(\bar{S}\cup(S'\cap\{i\}))+\mathbf{1}_{\{\bar{S}=\phi\wedge i\notin S'\}}+\widehat{\mathrm{Half}}_s(\bar{S}\setminus\{i\})\right)\mathbf{1}_{\{S\cup\{i\}\subseteq S'\}}$$

Using the fact that $|S'| < |S|$ therefore $\mathbf{1}_{\{S\cup\{i\}\subseteq S'\}} = 0$:

$$=\frac{1}{2}\widehat{\mathrm{Half}}_s(\bar{S}\cup(S'\cap\{i\}))\widehat{\mathrm{Maj}}_{n-s}((S\cup\{i\})\setminus S').$$

$\square$

Similar to the over-sparse setting, we say a neuron is *good* if $S' \subset S$, that is, if the selected variables are a subset of the relevant variables. Then we have,

**Lemma 20** (Population gradient at initialization for good neurons)**.** *Assuming $s < k$ with $s, k$ even, for good neurons, for $\bar{S} = S' \cap S$, we have*

$$\mathop{\mathbb{E}}_{\mathbf{x}\sim\{\pm 1\}^n}\left[\frac{\partial}{\partial w_i}h_{\mathbf{w}}(\mathbf{x})\cdot\chi_S(\mathbf{x})\right] = \begin{cases} \frac{1}{2}\widehat{\mathrm{Half}}_s(s)\widehat{\mathrm{Maj}}_{n-s}(k-s-1) & \text{if } i \in S\setminus S', \\ \frac{1}{2}\widehat{\mathrm{Half}}_s(s)\widehat{\mathrm{Maj}}_{n-s}(k-s+1) & \text{if } i \notin S, \\ 0 & \text{otherwise.} \end{cases}$$

*Proof.* Using Lemma 18 and 19, we get

1. $i \in S, i \in S'$:

$$\mathop{\mathbb{E}}_{\mathbf{x}\sim\{\pm 1\}^n}\left[\frac{\partial}{\partial w_i}h_{\mathbf{w}}(\mathbf{x})\cdot\chi_S(\mathbf{x})\right] = \frac{1}{2}\widehat{\mathrm{Half}}_s(\bar{S}\setminus\{i\})\widehat{\mathrm{Maj}}_{n-s}(S\setminus\bar{S})$$

2. $i \in S, i \notin S'$:

$$\mathop{\mathbb{E}}_{\mathbf{x}\sim\{\pm 1\}^n}\left[\frac{\partial}{\partial w_i}h_{\mathbf{w}}(\mathbf{x})\cdot\chi_S(\mathbf{x})\right] = \frac{1}{2}\widehat{\mathrm{Half}}_s(\bar{S})\widehat{\mathrm{Maj}}_{n-s}(S\setminus(\bar{S}\cup\{i\}))$$

3. $i \notin S, i \in S'$:

$$\mathop{\mathbb{E}}_{\mathbf{x}\sim\{\pm 1\}^n}\left[\frac{\partial}{\partial w_i}h_{\mathbf{w}}(\mathbf{x})\cdot\chi_S(\mathbf{x})\right] = \frac{1}{2}\widehat{\mathrm{Half}}_s(\bar{S}\cup\{i\})\widehat{\mathrm{Maj}}_{n-s}(S\setminus\bar{S})$$

4. $i \notin S, i \notin S'$:

$$\mathbb{E}_{\mathbf{x} \sim \{\pm 1\}^n} \left[ \frac{\partial}{\partial w_i} h_{\mathbf{w}}(\mathbf{x}) \cdot \chi_S(\mathbf{x}) \right] = \frac{1}{2} \widehat{\mathrm{Half}}_s(\bar{S}) \widehat{\mathrm{Maj}}_{n-s}((S \setminus \bar{S}) \cup \{i\})$$

Since $S' \subseteq S$, the arguments to the $\widehat{\mathrm{Half}}$ for (1) and (3) are odd. The corresponding Fourier coefficients for odd sets for Half are 0 (see O'Donnell (2014)), thus we have the desired result. □

**Theorem 21** (Formal version of Theorem 5). *Fix $s, k, \epsilon$ such that $s < k, \epsilon > 0$. Then for network width $r \geq \Omega((n/k)^s)$, the initialization scheme proposed here guarantees that for every $(n, k)$-parity distribution $\mathcal{D}$, with probability at least $0.99$ over the choice of sample and initialization, after one step of batch gradient descent with sample size $m = O((n/k)^{k-s-1})$ and appropriate choice of learning rate, there is a subnetwork in the one-layer ReLU MLP that has at least $\frac{1}{2} - \epsilon$ correlation with the parity function.*

*Proof.* To compute the parity function, we need to have $k$ neurons which identify the correct coordinates and have the appropriate biases. In order to identify the correct coordinates, we will focus only on good neurons, and on population gradient. We will then argue by standard concentration arguments that this holds from samples.

Let us consider a good neuron with $S' \subseteq S$. Firstly note that the scale of the bias is set such that it does not affect the gradient, Since it is at most $\epsilon/2$. Thus we can assume the no bias case for gradient computation. For all $i \in S \setminus S'$, that is, the set of relevant variables that are not selected in the initialization, let $\xi_{S \setminus S'}$ denote the gradient at initialization, and for all $i \notin S$, that is, the set of irrelevant variables, let $\xi_{[n] \setminus S}$ denote the gradient at initialization. Then using Lemma 20 and Lemma 17, we have

$$\frac{|\xi_{S \setminus S'}|}{|\xi_{[n] \setminus S}|} = \frac{n - s}{k - s - 1} > 1.$$

With $\eta$ and $\gamma$ being 0 on the bias terms and the second layer, $\eta = -\frac{\epsilon}{2k|\xi_{S \setminus S'}|}$ for the first layer weights, $\lambda = 1 - \frac{\epsilon}{2k}$ and $\gamma = |\xi_{[n] \setminus S}|$, one step of truncated population gradient descent gives us, for all $i \in S$, $w_i = \frac{\epsilon}{2k}$ and for all $i \notin S$, $|w_i| = \frac{\epsilon^2}{k}$. Since our initialization has two copies of the same neuron with second layer weights 1 and $-1$, one of them will have the gradient in the correct direction, ensuring the above, in particular, the one with the weight being $\mathrm{sign}(\xi_{S \setminus S'})$. Since $\epsilon$ can be set to be arbitrarily small, terms with $\epsilon^2$ can be ignored in comparison to terms with $\epsilon$. Thus the non-parity coefficients do not affect the output of the function and we get the appropriate parity coefficients (scaled by $\epsilon/2k$). To extend these guarantees to the batch gradient setting, we need to compute gradients to precision

$$\tau = |\xi_{[n] \setminus S}| / 2 = c_s (n - s)^{-\frac{k-s-1}{2}}$$

for some constant $c_s$ dependent only on $s$. This implies a sample complexity of $O(c_s^2 (n - s)^{k-s-1})$ using standard Chernoff bound. As for the width, we need to ensure that we have the required number of good neurons with appropriate bias. The probability of a randomly initialized neuron to be good is

$$\frac{\binom{k}{s}}{\binom{n}{s}} = \Theta \left( \left( \frac{k}{n} \right)^s \right).$$

The probability of choosing the appropriate bias is $1/(k + 1)$. Thus to be able to choose $k + 1$ good neurons with correct biases, we need width $O(k^2 (n/k)^s)$. □

We provide some additional remarks on the proof of Theorem 5:

- The sample complexity compared to the dense initialization studied by Barak et al. (2022) improves by a factor of $n^s$, at the cost of a higher width by a factor of $n^s$.

- We conjecture that Theorem 5 can be strengthened to an end-to-end guarantee for the full network, like Theorem 4. The technical challenge lies in analyzing how weight decay uniformly prunes the large irrelevant coordinates, without decaying the good subnetwork. We believe that this occurs robustly (from the experiments on sparse initialization), but requires a more refined analysis of the optimization trajectory.

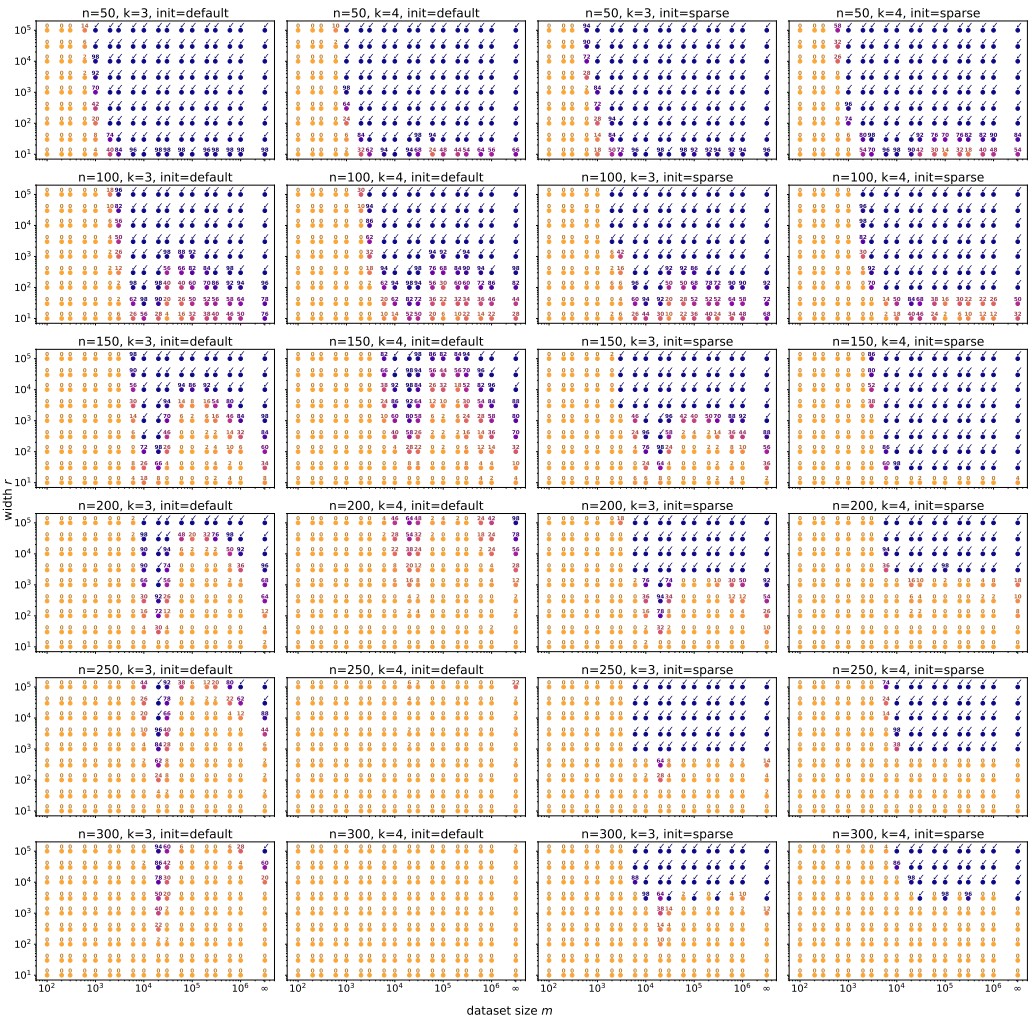

Figure 4: Full results for the sweep over dataset sizes and MLP widths. Each point represents 50 independent training runs of a 2-layer ReLU MLP on the offline sparse parity problem (input dimension $n$, parity degree $k$, initialization scheme, width $r$, dataset size $m$). All other algorithmic choices (learning rate $\eta = 0.1$, weight decay $\lambda = 0.01$, batch size $B = 32$, number of training iterations $T = 10^5$) are kept the same. We note the following: (1) **A success frontier**, where wide networks can learn at small sample sizes $m \ll BT$ (far outside online regime); (2) **Monotonic benefit of width**: overparameterization does not worsen the failure mode of overfitting in this setting, and amplifies success probabilities; (3) **Benefit of sparse axis-aligned initialization**: for sufficiently large $n$ where default initialization no longer works, sparse initialization scheme enlarges the feasible regime; (4) **Non-monotonic effects of dataset size:** there are unpredictable failures as we vary $m$ (horizontal slices of these plots).

## C  Full experimental results

### C.1  Full sweeps over dataset size and width

In our main set of large-scale synthetic experiments, we train a large number of 2-layer MLPs to solve various $(n, k)$-sparse parity problems, from $m$ samples. The full set of hyperparameters is listed below:

- Problem instance sizes: $n \in \{50, 100, 150, \ldots, 300\}$, $k \in \{3, 4\}$. Note that Barak et al. (2022) investigate empirical *computational time* scaling curves for larger $k$ (up to 8) in the online ($m = \infty$) setting. We are able to observe convergence for larger $k$ in the offline setting, but we omit these results from the systematic grid sweep (the feasible regime is too small in terms of $n$).

- Dataset size: $m \in \{100, 200, 300, 600, 1000, 2000, 3000, \ldots, 600000, 1000000\}$. We also include runs in the online regime (rightmost columns in Figure 4), which correspond to the regime studied by Barak et al. (2022).

- Network width: $r \in \{10, 30, 100, 300, \ldots, 10000, 30000, 100000\}$.

- Initialization scheme: PyTorch default (uniform on the interval $[-1/\sqrt{n}, 1/\sqrt{n}]$), and random 2-sparse rows. In coarser-grained hyperparameter searches, we found 2 to be the optimal sparsity constant for large-width ($\geq 1000$) regimes studied in this paper; we do not fully understand why this is the case. We also keep the PyTorch default initialization scheme (uniform on the width-dependent interval $[-1/\sqrt{r}, 1/\sqrt{r}]$) for the second layer, and use default-initialized biases.

At each point in this hyperparameter space, we conduct 50 training runs, and record the *success probability*, defined as the probability of achieving test error $\leq 10\%$ on a held-out sample of size $10^4$ within $T = 10^5$ training iterations. The hyperparameters for SGD, selected via coarse-grained hyperparameter search to optimize for convergence time in the $n = 200, k = 3, r = 10000$ setting, are as follows: minibatch size $B = 32$; learning rate $\eta = 0.1$; weight decay $\lambda = 0.01$.

Figure 4 summarizes all of our runs, and overviews all of the findings (1) through (4) enumerated in the main paper. We go into more detail below:

(1) **A "success frontier": large width can compensate for small datasets.** We observe convergence and perfect generalization when $m \ll n^k$. In such regimes, which are far outside the online setting considered by Barak et al. (2022), high-probability sample-efficient learning is enabled by large width. Note that neither our theoretical or empirical results have sufficient granularity to predict or measure the precise way the smallest feasible sample size $m$ scales with the other size parameters (like $n, k, r$). The theoretical upper bounds show that if $r = \Omega(n^k)$, idealized algorithms (modified for tractability of analysis) can obtain $O(\text{poly}(n))$ or even $O(\log(n))$ sample complexity, and smaller $r$ can yield milder reductions of the exponent.

(2) **Width is monotonically beneficial, and buys data, time, and luck.** Despite the capacity of wider neural networks to overfit larger datasets, we find that there are *monotonic sample-efficiency benefits* to increasing network width, in *all* of the hyperparameter settings considered in the grid sweep. This can be quickly quantitatively confirmed by starting at any point in Figure 4, and noting that success probabilities only increase[9] going upwards (increasing $r$, keeping all other parameters equal). Along some of these vertical slices, we observe that transitions from $0\%$ to $100\%$ are present: at these corresponding dataset sizes $m$, *large width makes sample-efficient learning possible*. Figure 2 (center) shows this in greater detail, by choosing a denser grid of sample sizes $m$ near the empirical statistical limit.

(3) **Sparse axis-aligned initialization buys data, time, and luck.** Used in conjunction with a wide network, we observe that a sparse, axis-aligned initialization scheme yields strong improvements on all of these axes. This can be seen by comparing the pairs of subplots in columns 1 vs. 3 and 2 vs. 4 in Figure 4. We found that $s = 2$ (i.e. initialize every hidden-layer neuron with a random 2-hot weight vector) works best for the settings considered in this study.

(4) **Intriguing effects of dataset size.** Unlike the monotonicity along vertical slices in Figure 4, some of the *horizontal* slices exhibit non-monotonic success probabilities. Namely, as $m$ increases, keeping all else the same, the network enters and exits a first feasible regime; then, at large enough sample sizes (including the online setting), learning is observed to succeed again. Sparse initialization reduces this counterintuitive behavior, but not entirely (see, e.g., the $n = 200, k = 3$ cell). We do not attempt to explain this phenomenon; however, we found in preliminary investigations that the locations of the transitions are sensitive to the choice of weight decay hyperparameter. Figure 2 (right) shows this in greater detail, plotting median convergence times (as defined above) instead of success probabilities.

---

[9]Small exceptions (such as $98\% \to 96\%$ for $n = 100, k = 4, m = 2000$) are all within the standard error margins of Bernoulli confidence intervals.

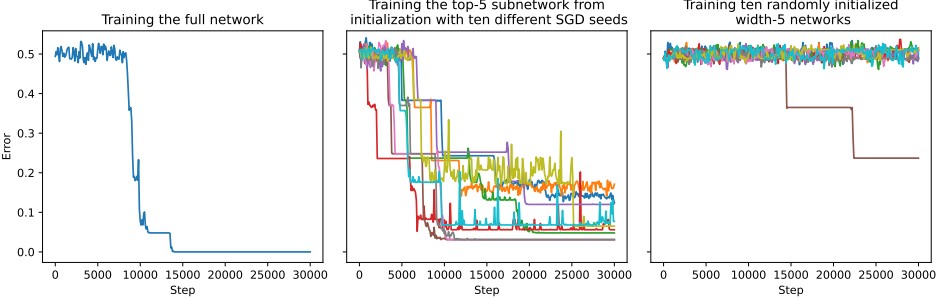

Figure 5: (Lottery tickets) Left: Training a width-100 MLP on the (n=50, k=5)-online sparse parity task, where each hidden neuron initialized with 2 non-zero incoming weights. Center: We prune all but the top 5 neurons by weight norm at the end of training; rewind weights to the original initialization, and retrain with various SGD random seeds. Right: The same as Center, but the weights are randomly reinitialized in each run.

## C.2 Lottery ticket subnetworks

Our theoretical analysis of sparse networks and experimental findings suggest that width provides a form of parallelization: wider networks have a higher probability of containing lucky neurons which have sufficient Fourier gaps at initialization to learn from the dataset. In Figure 5 we perform an experiment in the style of Frankle and Carbin (2018), showing that indeed the neurons which end up being important in a wide sparsely-initialized network form an unusually lucky 'winning ticket' subnetwork. When we rewind the weights of this subnetwork to initialization, its test error starts out poor, but when we train just this subnetwork from initialization its performance quickly improves, unlike the large majority of randomly initialized subnetworks of the same size. For this experiment, batch size=32 and learning rate=0.1.

## C.3 Training wide & sparsely-initialized MLPs on natural tabular data

As a preliminary investigation of whether our findings translate well to realistic settings, we conduct experiments on the tabular benchmark curated by Grinsztajn et al. (2022). For simplicity, we use all 16 numerical classification tasks from the benchmark. These datasets originate from diverse domains such as health care, finance, and experimental physics. Our main comparison is between MLPs (with algorithmic choices inspired by our sparse parity findings) and random forests Breiman (2001). We chose random forests as a baseline because they are known to achieve competitive performance with little tuning. Figure 6 summarizes our results, in which we find that wide and sparsely-initialized MLPs improve sample efficiency on small tabular datasets. We describe these experiments in full detail below.

**Data preprocessing.** We standardize the dataset, centering each feature to have mean 0 and normalizing each feature to have standard deviation 1.[10] For each task, we set aside 10% of the data for the test set, and downsample varying fractions of the remaining data to form a training set. We vary the downsampling fraction on an exponential grid: {1, 0.6, 0.3, 0.2, 0.1, 0.06, 0.03, 0.02, 0.01, 0.006, 0.003, 0.002, 0.001, 0.0006, 0.0003, 0.0002, 0.0001}. In each of 100 i.i.d. trials for each setting, we re-randomize the validation split as well as the downsampled training set.

**Algorithms.** MLPs are trained using the same architectural choices noted in Section C.1, except the hyperparameters noted below. We use the scikit-learn Pedregosa et al. (2011) RandomForestClassifier. Hyperparameters not mentioned are set to library defaults.

**Hyperparameter choices for MLPs.**

---

[10]Note that Grinsztajn et al. (2022) instead transform each feature such that its empirical marginal distribution approximates a standard Gaussian.

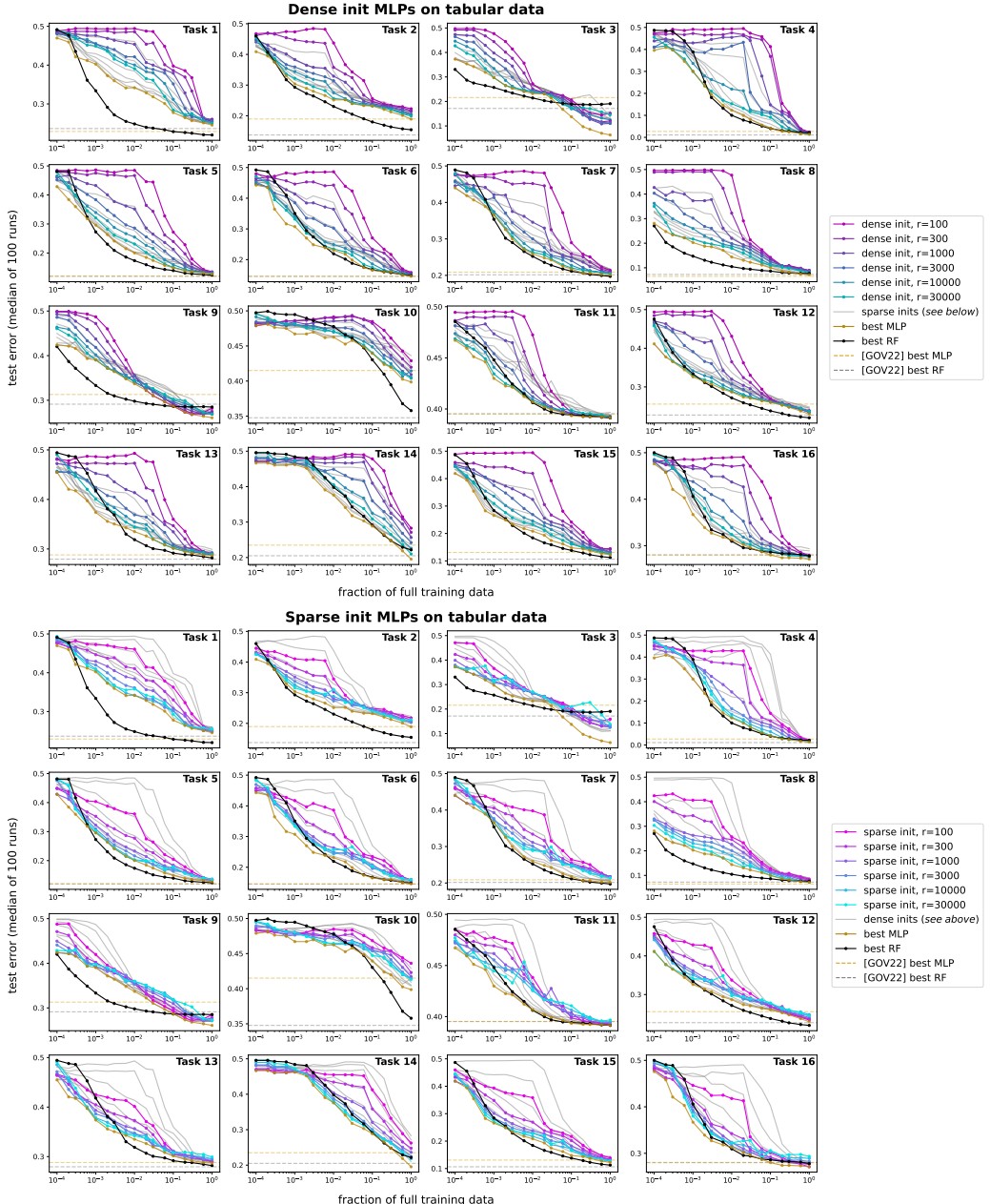

Figure 6: Full results of our tabular data experiments (following Grinsztajn et al. (2022)), varying MLP width $r$, sample size $m$ (via downsampling the training data), and initialization sparsity. Both width and sparsity tend to improve generalization, especially for small datasets. See Tables 1 and 2 for further details.

| Task | OpenML identifier | # features | # examples |
|------|------------------|-----------|-----------|
| 1 | credit | 11 | 16714 |
| 2 | electricity | 8 | 38474 |
| 3 | covertype | 11 | 566602 |
| 4 | pol | 27 | 10082 |
| 5 | house_16H | 17 | 13488 |
| 6 | MagicTelescope | 11 | 13376 |
| 7 | bank-marketing | 8 | 10578 |
| 8 | MiniBooNE | 51 | 72998 |
| 9 | Higgs | 25 | 940160 |
| 10 | eye_movements | 21 | 7608 |
| 11 | Diabetes130US | 8 | 71090 |
| 12 | jannis | 55 | 57580 |
| 13 | default-of-credit-card-clients | 21 | 13272 |
| 14 | Bioresponse | 420 | 3434 |
| 15 | california | 9 | 20634 |
| 16 | heloc | 23 | 10000 |

Table 1: Metadata for the 16 tabular classification benchmarks, curated by Grinsztajn et al. (2022) (January 2023 version, benchmark suite ID 337) and publicly available via OpenML.

- Width $r$: $\{100, 300, 1000, 3000, 10000, 30000\}$. For deeper networks, the hidden layers are set to $r \times r$.

- Depth (number of hidden layers): $\{1, 2, 3\}$. In all of these settings, depth-1 networks are nearly uniformly outperformed by deeper ones of the same width.

- Sparsity of initialization: $s \in \{2, 4\}$, and also dense (uniform) initialization. 4-sparse initialization is nearly uniformly outperformed by 2-sparse (in agreement with our experiments on sparse parity).

- Weight decay: $\{0, 0.01\}$. In these settings, this hyperparameter had a minimal effect.

- Learning rate: $10^{-3}$

- Batch size: $256$

- Training epochs: $100$

**Hyperparameter choices for RandomForestClassifier.**

- `max_depth`: $\{-1, 2, 3, 4, 10\}$

- `max_features`: $\{1, 2, 3, 4, \mathrm{sqrt}, \mathrm{None}\}$

- `n_estimators`: $\{10, 100, 1000\}$

**Larger widths.** For all of the tasks except 2 and 9 (since these are significantly larger datasets), we include extra runs with even larger networks: depth-2 MLPs with non-uniform width (i.e. sequence of hidden layer dimensions) $\{(100000, 10000), (10000, 100000)\}$.

**Plots in Figure 6.** For clarity of presentation, in the plots where we vary MLP width and initialization sparsity, we fix depth to 3 and sparsity level $s = 2$. Qualitative trends are similar for other settings. The gold "best MLP" curves show the best median-of-100 validation losses across all architectures in the search space.

**Comparison with full-data baselines.** To ensure that our baseline algorithm choices for MLPs and random forests are reasonable, we present a comparison with the results of Grinsztajn et al. (2022) (who performed extensive hyperparameter search) on the full datasets. These are shown as the dotted lines in Figure 6, as well as Table 2.

| Task | Best MLP | | | Best RF | | | (Grinsztajn et al., 2022) | | |
|---|---|---|---|---|---|---|---|---|---|
| | full | 10% | 1% | full | 10% | 1% | MLP | RF | Best model |
| 1 | 24.5 | 27.8 | 34.2 | **22.0** | **22.9** | **24.8** | 22.9 | 23.6 | 22.5 (GBT) |
| 2 | 18.9 | 23.2 | 25.1 | **15.4** | **18.0** | **23.0** | 23.6 | 13.7 | 12.8 (XGB) |
| 3 | **6.3** | **13.7** | 23.2 | 19.1 | 18.9 | **21.4** | 21.6 | 17.1 | 17.1 (RF) |
| 4 | **1.4** | 4.2 | 12.3 | 2.1 | **4.0** | **10.0** | 6.3 | 1.9 | 1.7 (XGB) |
| 5 | 12.7 | 14.9 | 20.0 | **12.4** | **13.9** | **17.4** | 12.1 | 12.0 | 11.1 (XGB) |
| 6 | **14.6** | **16.5** | 23.1 | 14.8 | 16.8 | **21.9** | 14.6 | 14.5 | 13.9 (FTT) |
| 7 | 20.2 | 21.8 | 25.9 | **19.7** | **21.1** | **25.2** | 20.9 | 20.1 | 19.5 (GBT) |
| 8 | **7.2** | 10.0 | 14.4 | 7.7 | **8.6** | **10.5** | 6.7 | 7.3 | 6.2 (XGB) |
| 9 | **26.1** | **28.7** | 33.7 | 28.5 | **28.7** | **29.9** | 31.4 | 29.1 | 28.6 (XGB) |
| 10 | 39.9 | 43.9 | **46.3** | 35.8 | 43.0 | 47.8 | 41.8 | 34.8 | 33.4 (XGB) |
| 11 | **39.1** | **39.3** | 40.7 | **39.1** | 39.4 | **40.6** | 39.5 | 39.5 | 39.4 (XGB) |
| 12 | 23.0 | 26.1 | 28.7 | **22.0** | **23.8** | **27.2** | 25.5 | 22.7 | 22.0 (XGB) |
| 13 | 28.7 | 30.1 | 33.5 | **28.2** | **29.2** | **31.9** | 28.9 | 28.0 | 28.0 (RF) |
| 14 | **19.5** | **28.9** | **37.6** | 22.2 | 29.2 | 39.8 | 23.4 | 20.5 | 20.5 (RF) |
| 15 | 12.2 | 14.8 | 20.8 | **11.2** | **13.8** | **18.4** | 12.9 | 10.6 | 9.7 (XGB) |
| 16 | **27.1** | **28.0** | 31.3 | 27.8 | 28.6 | **31.0** | 27.8 | 28.1 | 27.4 (ResNet) |

Table 2: Numerical comparisons for the tabular data experiments, to accompany Figure 6. We report the median test error (%) over 100 i.i.d. training runs of MLPs and random forests. The 3 subcolumns in each group denote models trained on the full dataset and their {10%, 1%} downsampled counterparts. For all of these results, bootstrap 95% confidence intervals have width < 2%. We observe that wide and/or sparsely-initialized MLPs are competitive with tree-based methods. In the rightmost 3 columns, we provide test errors for the same tasks, reported by (Grinsztajn et al., 2022) (GBT = gradient boosted tree, XGB = XGBoost, FTT = Feature Tokenizer + Transformer (Gorishniy et al., 2021)). Note that our cross-validation protocol differs slightly (to handle variance incurred by downsampling), which may explain performance discrepancies. We include these only to illustrate that our full-data accuracies are commensurate with those in prior works focused exclusively on benchmarking models for tabular data.

**Results.** We note the following findings from the results in Figure 6 and Table 2:

(2T) **Wide networks resist overfitting on small tabular datasets.** Like in the synthetic experiments, width yields monotonic end-to-end benefits for sample-efficient learning on these datasets. This suggests that the "parallel search + pruning" mechanisms analyzed in our paper are empirically at play in these settings, and that these networks' capacity to overfit does not preclude nontrivial feature learning and generalization. These comparisons can be seen by comparing the colored curves (which represent depths 2 and 3) within each subplot in Figure 6. In some (but not all) cases, these MLPs perform competitively with hyperparameter-tuned random forest classifiers.

(3T) **Sparse axis-aligned initialization sometimes improves end-to-end performance.** These comparisons can be seen by comparing vertically adjacent sparse vs. dense subplots in Figure 6. This effect is especially pronounced on datasets which are downsampled to be orders of magnitude smaller. We believe that this class of drop-in replacements for standard initialization merits further investigation, and may contribute to closing the remaining performance gap between deep learning and tree ensembles on small tabular datasets.

## C.4 Software, compute infrastructure, and resource costs

GPU-accelerated training and evaluation pipelines were implemented in PyTorch (Paszke et al., 2019). Each training run was performed on one GPU in an internal cluster, with NVIDIA P40, P100, V100, and RTX A6000 GPUs. A single $T = 10^5$ training run took 10 seconds on average (with early termination for the vast majority of grid sweeps); across all of the results in this paper, around $2 \times 10^5$ training runs were performed, in a total of 600 GPU-hours. Note that the precise evaluation of test error (at batch size $10^4$) constitutes a significant portion of the computational cost; this necessitated mindful GPU parallelization.