# OpenReview forum: "Pareto Frontiers in Deep Feature Learning: Data, Compute, Width, and Luck"
_NeurIPS.cc/2023/Conference — NeurIPS 2023 spotlight_

### Official Review · Reviewer_9731 · 2023-07-07

**Soundness:** 3 good
**Presentation:** 3 good
**Contribution:** 3 good
**Rating:** 7
**Confidence:** 3

**Summary:**

This paper considers learning parity functions with neural networks and particularly studies the tradeoff between size of the network and the sample size. The paper also explore a connection between the network size the lottery ticket hypothesis. The neural network studied in this paper has a sparse, specific, and symmetric structure. One step gradient arguments under the hinge loss have been used for the training. Finally, the paper proposes a promising direction for processing tabular datasets using neural networks.

**Strengths:**

- The paper shows that with sparse initialization, one can decide between wide network and smaller number of samples, or narrower network and higher number of samples and basically move between the two regime.
- The connection of width to the lottery ticket hypothesis in the sparse initialization setting.
- The experiments are quite extensive for the 2-layer MLP model with sparse initialization.
- Observation of grokking and sample-wise double descent in the experiments is also interesting. This shows a potential avenue for future theoretical research on grokking.

**Weaknesses:**

- In the abstract, the lower-bound is stated for the number of training samples. However, the lower bound with SQ only contains the gradient precision. (Large number of samples gives is sufficient for the condition on gradient precision but not necessary). So to be exact, there is no bound on the number of samples. Moreover, the SQ bound does not give information about the SGD algorithm.
- It would have been great if there were experiments checking the results of this paper beyond that particular setting of theoretical results. For example, it would have been nice to have the same experiments on Transformers and mean-field networks. Also the batch size is always 32 in the experiments; it would be nice to also try larger and smaller (maybe one sample at time) batch-sizes as well. Further, what happens if we do not have the initialization sparsity?

**Questions:**

The setting considered in the paper is quite specific (e.g., the initialization and the training of the network). So is it possible to theoretically/empirically discuss the limitations of this study. For example if we keep the parity function as the target, do we expect these result to generalize to other architecture with more standard training methods?

Suggestion:\
The appendix part for the proof of the positive part is a bit hard to read. Particularly, if would be great if the exact training algorithm is once explained and also there is an overview over the proof.

**Limitations:**

This is a theoretical study and there is no negative societal impact. However, the potential limitations of the findings could be discussed more extensively.

---

> ### Author Rebuttal · Authors · 2023-08-10
>
> Thanks for the feedback and suggestions!
>
> **(W1) Gap between SQ and SGD**: We agree with the reviewer that the SQ framework as stated only gives a lower bound dependent on total number of queries and the precision of the queries. This does not directly imply a lower bound on the sample complexity, and is a known gap between SQ and SGD. We note that there has been work extending the SQ model to the honest SQ model (HSQ) [Yang05] which evaluates each query by sampling $M$ samples independently and providing the empirical average as the answer to the query. In these settings, tolerance is replaced by the number of samples and there are results of the following style: SQ dimension $d$ implies a lower bound on the total number of samples used $\Omega(d/\log d)$. We will add a discussion about this in the paper.
>
> _[Yang05] Ke Yang. New lower bounds for statistical query learning. J. Comput. Syst. Sci., 2005._
>
> **(W2, Q1) Other settings**: We thought the more appropriate direction of "beyond theory" was to go from our synthetic parity setting to real tabular data. For the parity problem in particular, the effect of choice of architectures has been previously explored in [BEGKMZ‘22], and we believe similar results may hold here as well. As for initializations beyond sparsity, our experiments show similar results to the sparse initialization setting. However it is technically significantly challenging to analyze since it boils down to showing anti-concentration of higher-order Fourier coefficients of random halfspaces, which we do not currently know how to do. (The experiments in Appendix C.1 of [BEGKMZ22] provide empirical evidence that these halfspaces’ Fourier spectra behave like those of majority with high probability.)
>
> _[BEGKMZ22] Boaz Barak, Benjamin Edelman, Surbhi Goel, Sham Kakade, Eran Malach, and Cyril Zhang. "Hidden progress in deep learning: Sgd learns parities near the computational limit." NeurIPS 2022._
>
> **(S1) Clarity of proofs in appendix**: We appreciate the feedback, and will add high-level intuitive sketches to the main paper and the appendix.

---

> > ### Comment · Reviewer_9731 · 2023-08-16
> >
> > Thank you for the clarifications. Indeed adding more detail on the sample complexity bounds and potential limitations (e.g., batch size and deriving lower bounds) would be very nice. I will maintain my score.

---

### Official Review · Reviewer_qvgd · 2023-07-07

**Soundness:** 4 excellent
**Presentation:** 4 excellent
**Contribution:** 4 excellent
**Rating:** 8
**Confidence:** 4

**Summary:**

In this paper, the authors study the tradeoff between various resources in feature learning---data, compute, width. The authors focus on the fundamental problem of learning parity functions using a two-layer MLP. The degree $k$ of a parity function of $n$ variables controls the hardness of the problem: generally a bigger $k$ means more difficult. Learning parity function is a fundamental problem which has been used to study deep learning in the literature. The authors consider learning $(n,k)$-parities using gradient descent (GD) sparsely initialized at sparsity level $s$.

The first main result shows that if $s > k$ (over-sparse initialization), then there is a tradeoff between network width $r$ and sample size $m$ required to learn a $(n,k)$-parity with high probability. This result interpolates existing feature-learning results as far as I know. The second main result shows that if $s<k$ (under-sparse initialization), there is a similar width vs. sample size tradeoff for one-step GD.

Extensive simulations and experiments on natural tabular datasets are presented to complement the theory.


**Strengths:**

I find this paper impressive and enjoyable to read. Understanding the tradeoff of different resources (time, memory, data) is important for large-scale machine learning. Focusing on the fundamental problem of learning parity functions, the authors present solid theoretical results. I think both theoretical and empirical contributions are significant.

The theory also provides perspectives for the lottery ticket hypothesis, by showing that a large width is beneficial for finding the right "lottery ticket".

The writing is very clear. High-level ideas are well explained and figures are easy to understand.

**Weaknesses:**

I cannot evaluate the technical novelty of this paper as I do not claim expertise in learning parity functions, while some of the analytical strategies seem to be based on existing papers.

One minor comment is that theory is established for shallow networks, and it cannot reflect "Deep Feature Learning" in the title.



**Questions:**

I am curious (1) if the authors could comment on the role of depth in neural networks, and (2) how do the results connect to the staircase phenomenon?

---

> ### Author Rebuttal · Authors · 2023-08-10
>
> We thank the reviewer for their positive feedback.
>
> **(W1) “Deep feature learning” terminology**: Our intent for the shorthand “deep feature learning” in the title was to refer to gradient-based feature learning in neural networks, as opposed to the NTK regime. We understand the possible unintended interpretation of “deep” as “containing a many-layer hierarchy” (which we believe are currently clarified in the abstract and main body), and are considering minor adjustments to the title.
>
> **(Q1) Role of depth**: How depth $>2$ changes the picture sketched in this paper is non-obvious due to the entanglements with optimization, and is an excellent direction for future work especially for tasks that (unlike the parity function) have hierarchical structure. For our paper, we focused on depth-2 because it is more tractable to analyze but still not well-understood, and the number of hyperparameter knobs we analyzed was expansive even without adding depth.
>
> **(Q2) Relation to staircase phenomenon**: Compared to the original staircase paper [ABBBN21]  where each staircase added a single variable, the sample/runtime is already polynomial in $n$ therefore this trade-off is not super interesting. In the more recent paper [ABM23] which quantifies the leap complexity, we speculate that the finite-sample tradeoffs we describe may apply to each leap in a staircase. This could be an interesting subject for future work.
>
> _[ABBBN21] Emmanuel Abbe, Enric Boix-Adsera, Matthew S. Brennan, Guy Bresler, and Dheeraj Nagaraj. The staircase property: How hierarchical structure can guide deep learning. NeurIPS 2021._
>
> _[ABM23] Emmanuel Abbe, Enric Boix-Adserà, and Theodor Misiakiewicz. SGD learning on neural networks: leap complexity and saddle-to-saddle dynamics. COLT 2023._

---

### Official Review · Reviewer_T64v · 2023-07-07

**Soundness:** 3 good
**Presentation:** 3 good
**Contribution:** 3 good
**Rating:** 5
**Confidence:** 1

**Summary:**

Disclaimer: My expertise in this domain is limited and understanding is highly superficial.

The manuscript presents a detailed take on addressing the tradeoffs on 4 axes, namely model size, dataset size, training epochs, and stochasticity. It offers a well-rounded analysis of the area and provides empirical and theoretical evidence for intuitive ideas, increasing network width leading to better sample efficiency. They use sparse parity learning as a proxy for real world feature learning problems which might be aligned on the above mentioned 4-axes. They also provide curious insights like the network width improves the possibility of “winning lottery ticket” neurons. The wide, sparsely initialized MLPs also outperform the very strong RFs.

**Strengths:**

- Section 4.1 and C.1: The statements on interplay between width and dataset size for example are a good starting point for further focused research.
- Section B: Detailed theoretical contributions in the supplementary section verify the experiments and intuition.
- Highly exhaustive experiments, analyzing and investigating important questions.

**Weaknesses:**

- The paper, while having a good base, introduction and motivation, becomes hard to follow and the organization seems a bit unnatural. For example, in my opinion, moving the insight from Lines 823 to 853 to the main paper at the expense of the rather verbose Section 3.2, before the “actual” analysis starts in Line 223 might help pitch the paper better.
- It’s not clear to me how this work is different from the rich vein of work in learning parities and providing a SQ learning bound. It might be worthwhile adding a section comparing against previous attempts and drawing focus to distinct contributions, or the work might seem incremental in several directions without making a significant difference in one direction.

**Questions:**

- See Weaknesses.
- How is the work distinct from the previous works like [1], especially as theorem 4 seems like a trivial extension of informal theorems and analysis from [1]? A bit more formalization of the distinction would be helpful.
- More comparisons on the nC2 combinations of data-width-time-luck would provide one-on-one insights on the interplay and could be a good addition to the work. Almost a matrix of sorts, highlighting the relationship between two axes and their effect overall on the network could summarize the findings well.

[1] Barak, Boaz, et al. "Hidden progress in deep learning: Sgd learns parities near the computational limit." Advances in Neural Information Processing Systems 35 (2022): 21750-21764.

---

> ### Author Rebuttal · Authors · 2023-08-10
>
> Thanks for the thoughtful comments and questions.
>
> **(W1) Paper organization**: Thanks for the feedback and suggestions. We will improve the presentation, and add intuitive overviews of the technical proofs.
>
> **(W2, Q2) Relation to other work on NN parity learning**: We have not seen following contributions in prior work on SQ parity learning with NNs: (1) a “success frontier”-- a parametric family of algorithms which trade off the heterogeneous resources of data & computation (2) experiments which corroborate these tradeoffs. We will add a more detailed quantitative comparison to Appendix A.1. The main points are below:
> - [BEGKMZ22] shows that one step of gradient descent on a single neuron is able to recover the indices corresponding to the parity with $n^{O(k)}$ samples/computation. The present work expands on this with the following: (1) extends consideration to the finite-sample setting, not just online learning, where the heterogeneous resource tradeoff frontier arises; (2) shows how increasing width improves sample efficiency; (3) introduces sparsity of initialization as a hyperparameter, which has interplay with width & sample-efficiency.
> - [ABM23] improves this bound to $O(n^{k-1})$ online SGD steps and generalizes the result to handle hierarchical staircases of parity functions which requires a multi-step analysis.
> - [Telgarsky23] studies the problem of 2-sparse parities with two-layer neural networks trained with vanilla SGD (unlike our restricted two-step training algorithm) and studies the margins achieved post training. They use the margins to get optimal sample complexity $\tilde{O}(n^2/\epsilon)$ in the NTK regime. Going beyond NTK, they analyze gradient flow (with certain additional modifications) on an exponentially ($n^n$) wide 2-layer network (making it computationally inefficient) to get the improved sample complexity of $\tilde{O}(n/\epsilon)$.
> In contrast to these works, our theoretical contribution (Theorem 4) highlights the improvement in sample complexity while maintaining computational efficiency which we achieve using the sparse initialization.
>
> _[BEGKMZ22] Boaz Barak, Benjamin Edelman, Surbhi Goel, Sham Kakade, Eran Malach, and Cyril Zhang. "Hidden progress in deep learning: Sgd learns parities near the computational limit." Advances in Neural Information Processing Systems 35 (2022): 21750-21764._
>
> _[ABM23] Emmanuel Abbe, Enric Boix-Adserà, and Theodor Misiakiewicz. SGD learning on neural networks: leap complexity and saddle-to-saddle dynamics. COLT 2023._
>
> _[Telgarsky23] Matus Telgarsky. Feature selection and low test error in shallow low-rotation ReLU networks. ICLR 2023._
>
> **(Q3) Additional pairwise tradeoffs**: We will attempt to provide further visualizations to elucidate the “${n \choose 2}$ grid” of pairwise resource tradeoffs in the revision. There are some methodological considerations (high sensitivity to the choice of accuracy threshold; effect of optimizer hyperparameters and initialization scales on the precise convergence time, …) which make some of these comparisons hard to make quantitatively.

---

### Official Review · Reviewer_XFsj · 2023-07-12

**Soundness:** 3 good
**Presentation:** 4 excellent
**Contribution:** 3 good
**Rating:** 5
**Confidence:** 4

**Summary:**

In an attempt to explore the mechanisms behind generalization and training of neural networks and different elements that play a role in it,  this paper investigates the impact of four resources of training MLPs on the famous and well-studied (n, k)-parity learning problem. The authors conduct a massive grid of experiments to analyze the trade-offs between data, compute, width (model size) and luck (sparsity at initialization) when training MLPs on the mentioned problem. The experimental evaluation suggests existence of "frontiers" of resource trade-offs such that decrease any resource at the frontier would likely result in incomplete or non-existent generalization. They further theoretically analyze this frontier and prove that under the necessary assumptions, this frontier can be recovered (at least in the over-sparse initialization case) theoretically. Based on the observed patterns, the authors show that under correct combination of resources, neural networks are capable of achieving similar or in some cases superior performance than tree-based methods on small tabular tasks.

**Strengths:**

* The diverse and large-scale experiments show clear patterns of existence of the trade-offs proposed by the authors, and provide thorough evaluation of different combinations of the four mentioned resources.
* The theoretical results are supportive of the patterns present in the experiments and agree with the results from recent work.
* Although the presented results may not seem novel, and arguably have been speculated before, a study that extensively studies the concept of "success frontier" was lacking.
* This work addresses an important aspect of using neural networks in practice: tabular data and suggests that further improvements can be made in applying neural networks on these kind of problems. This is in contrary to the popular belief that neural networks are inferior than tree-based models on these datasets.
* The manuscript is well-presented, and to my knowledge, the authors have covered most related work and included necessary discussion and comparison with related work, which helps in conveying the importance of this setup and the findings.

**Weaknesses:**

I believe that this work has the potential to be very influential, but there are still some aspects that should be improved and some points that could be better justified.

### Major concerns:
* It's not clear if the study would apply to other problems or not. Hence, claiming that the results apply to "deep feature learning in general" sounds very exaggerated. It is definitely intuitive that the observed phenomena should extend to other problems to some extent, but we can't claim anything just based on intuition. For instance, it has been observed that width is not necessarily monotonically beneficial in all problems, and this could apply to other resources discussed in the paper as well. As a suggestion, a few ablation studies of smaller scale on other problems could be a better support for hypothesizing that the observations convey to other settings as well.
* Section 3.1: Correct me if I'm getting this wrong, but based on Proposition 3 it seems like there "exists" settings in which SGD can't find the correct solution. If I'm right, this doesn't mean "GD will fail" at all, and it just means that GD is not guaranteed to converge.
* In Theorem 4, setting error close to zero would result in $s \to \infty$, and $r \to 0$. How is this possible? Isn't $r$ required to be at least as large as $s$? Could the authors clarify the setting of this theorem?
* I'm not convinced about some references made to the lottery ticket hypothesis and "good neurons", and that width buys "Luck" in general. Although LTH could be a plausible explanation for the observations in the experiments, totally attributing the affect of width to LTH and "good neurons" would require thorough evidence, may it be theoretical or empirical.

### Minor issues:
* The authors have mentioned that they use bootsrapping for neural nets to solve the low-data problem. Have the authors used bootstrapping for other models that don't inherently perform it to have a fair comparison?
* In theorem 5, and theorem 21 (the formal version), it is mentioned that the problem will be solved "approximately". Can the authors please clarify this approximation?

**Questions:**

I have mentioned my questions in the "Weaknesses" section.

**Limitations:**

The authors have discussed the limitations.

---

> ### Author Rebuttal · Authors · 2023-08-10
>
> We thank the reviewer for their thoughtful review. Here we address the main concerns raised by the reviewer.
>
> **(W1) “Deep feature learning” terminology**: (Copied from response to R3) Our intent for the shorthand “deep feature learning” in the title was to refer to gradient-based feature learning in neural networks, as opposed to the NTK regime. We understand the possible unintended interpretation of “deep” as “containing a many-layer hierarchy” (which we believe are currently clarified in the abstract and main body), and are considering minor adjustments to the title.
>
> **(W2) Quantifiers in Proposition 3**: Proposition 3 is phrased according to the standard convention in learning theory that learning a concept class requires probable success for all concepts in the class. The proof can easily be tweaked to show that the same result applies for a $1-\epsilon$ fraction of all $(n,k)$-parities, if $\frac{rT}{\tau^2 \delta} \leq \frac{\epsilon}{2}\binom{n}{k}$.
>
> **(W3) $\epsilon$ in Theorem 4**: The statement of Theorem 4 only makes sense when $\epsilon > c_1/\sqrt{n}$, since sparsity $s$ cannot be greater than $n$.
>
> **(W4) Lottery ticket hypothesis**: We agree that our results and experiments establish that width buys “luck” in our specific setting, not necessarily in general. We will make the connection to LTH clearer in the revision.
>
> **(W5) Clarification on “bootstrapping”**: By “bootstrapping” the classification benchmark, we mean randomly downsampling the training data, inducing a distribution over harder supervised learning tasks. These smaller-sample tasks are not present in [GOV22]; hence, their reported numbers are only comparable with the rightmost points of the “error vs. sample size” curves. We will adjust the wording to minimize confusion.
>
> _[GOV22] Grinsztajn, Léo, Edouard Oyallon, and Gaël Varoquaux. "Why do tree-based models still outperform deep learning on typical tabular data?." NeurIPS 2022._
>
> **(W6) Clarification on “approximate solution”**: As is standard in learning theory, approximation refers to the learned solution being $\epsilon$-close in terms of error to the optimal solution.

---

### Decision · Program_Chairs · 2023-09-21

**Decision:**

Accept (spotlight)

**Comment:**

This paper presents a theoretical and empirical study of the tradeoffs between computational resources and statistical efficiency in deep learning. The authors investigate the performance gap between neural networks and tree-based classifiers on tabular data and identify three common difficulties for neural networks in this domain. They also explore the benefits of increasing network width and using sparse axis-aligned initialization in sparse parity learning.

The main strengths of this paper include its clear presentation of theoretical and empirical results, its thorough exploration of algorithm design choices in deep learning, and its practical insights for improving the performance of neural networks on tabular data. The authors provide a standardized suite of classification benchmarks with numerical input features, which can be used to evaluate the performance of different algorithms on tabular data. They also offer practical recommendations for improving the performance of neural networks on sparse parity learning tasks. Overall, this paper provides valuable insights for researchers and practitioners working in the field of (gradient-based) feature learning in neural networks.

Overall, this paper is a joy to read, and all four experts unanimously support this paper. While a few concerns were initially raised, they were subsequently clarified by authors during the rebuttal period. Overall, AC sees this paper as a clear accept case and a noteworthy NeurIPS contribution.